# Menin directs regionalized decidual transformation through epigenetically setting PTX3 to balance FGF and BMP signaling

Mengying Liu [1,2,4], Wenbo Deng [2,4], Lu Tang[2], Meng Liu[1], Haili Bao[2], Chuanhui Guo[2], Changxian Zhang [3], Jinhua Lu [2], Haibin Wang [2✉], Zhongxian Lu [1,2✉] & Shuangbo Kong [2✉]

During decidualization in rodents, uterine stroma undergoes extensive reprograming into distinct cells, forming the discrete regions defined as the primary decidual zone (PDZ), the secondary decidual zone (SDZ) and the layer of undifferentiated stromal cells respectively. Here we show that uterine deletion of *Men1*, a member of the histone H3K4 methyltransferase complex, disrupts the terminal differentiation of stroma, resulting in chaotic decidualization and pregnancy failure. Genome-wide epigenetic profile reveals that *Men1* binding in chromatin recapitulates H3K4me3 distribution. Further transcriptomic investigation demonstrates that *Men1* directly regulates the expression of PTX3, an extra-cellular trap for FGF2 in decidual cells. Decreased *Ptx3* upon *Men1* ablation leads to aberrant activation of ERK1/2 in the SDZ due to the unrestrained FGF2 signal emanated from undifferentiated stromal cells, which blunt BMP2 induction and decidualization. In brief, our study provides genetic and molecular mechanisms for epigenetic rewiring mediated decidual regionalization by *Men1* and sheds new light on pregnancy maintenance.

[1] School of Pharmaceutical Sciences, State Key Laboratory of Cellular Stress Biology, Xiamen University, Xiamen, Fujian, China. [2] Fujian Provincial Key Laboratory of Reproductive Health Research, Department of Obstetrics and Gynecology, The First Affiliated Hospital of Xiamen University, School of Medicine, Xiamen University, Xiamen, Fujian, China. [3] Centre de Recherche en Cancérologie de Lyon, Université Lyon 1, Inserm U1052, CNRS UMR5286, Lyon F-69000, France. [4] These authors contributed equally: Mengying Liu, Wenbo Deng. ✉email: haibin.wang@vip.163.com; zhongxian@xmu.edu.cn; shuangbo_kong@163.com

Successful pregnancy requires rapid remodeling of the endometrium in a process termed decidualization, which requires the blastocyst attachment as a trigger and occurs in a region-specific manner in mice[1]. At the initiation of decidualization, stromal cells surrounding the implanted blastocyst undergo proliferation and postmitotic differentiation extensively, forming the primary decidual zone (PDZ) between day 5 (day 1 = the day of positive of vaginal plug) and day 6 of pregnancy[2]. Subsequently, stromal cells adjacent to the PDZ continue to proliferate and differentiate into decidual cells to form the secondary decidual zone (SDZ) at the antimesometrial (AM) pole by day 8. A large portion of decidual cells experience polyploidization, characterized by large mono or bi-nucleated cells that undergo repeated DNA replication without cytokinesis[3]. In addition, a thin layer of undifferentiated stromal cells lines between the myometrium and the SDZ[4]. It is believed that the regionalized decidual structure is not only important for the supply of nutrition to the developing embryo but also acts as a barrier against excessive invasion of trophoblast cells[1,5]. While only small incrementation has been made about the underlying mechanism directing programmed stromal cells differentiation in recent decades.

Several studies have demonstrated that a set of central genes in evolutionarily conserved signaling pathways during embryogenesis, such as BMP (bone morphogenetic protein), WNT (wingless-type MMTV integration site family), FGF (fibroblast growth factor) signaling, and HOX (homeobox) genes, possess the potential function in building the regional pattern of decidualization[4,6,7]. BMP2, an indispensable factor for the differentiation of stromal cells into decidual cells, is intensively expressed in PDZ and SDZ[6,8]. The expression of WNT ligand Wnt4, which is also critical for uterine stromal cell differentiation, largely overlapped with Bmp2[4]. Meanwhile, Sfrp4, a WNT antagonist, is observed in the layer of undecidualized stromal cells outside the SDZ[4]. In addition to Sfrp4, Fgf2, a member of FGF family, also exhibits a similar expression pattern[6]. The fine-tuned interplay of these regionalization determinants synergistically or antagonistically promotes tissue homeostasis during decidualization. However, how these genes are coordinated to specify the allocation of decidual cells remains largely unknown.

Menin, encoded by the Men1, is a multifunctional scaffold protein participating in a variety of cellular functions, whose deficiency contributes to embryonic lethality at E11.5-E13.5 with defects in multiple organs[9,10]. Menin regulates target genes expression in a cell type-specific manner, including Hox genes, Cdkn1b, and Cdkn2c, Esr1 and p35, by interacting with chromatin-modifying enzymes, in particular, the histone methyltransferase (HMT) MLL1/2 that catalyze histone 3 lysine 4 trimethylation (H3K4me3)[11–15]. Regarding that early pregnancy is a precisely regulated process, whether the target genes of Menin-MLL complex and how Menin and its interplay with H3K4me3 modification participate in embryo implantation and decidualization remain elusive.

In the current study, we utilize a combination of genetic, biochemical, and pharmacological approaches to investigate the role of Men1 in peri-implantation uteri and reveal that Men1 ablation disrupts the appropriated differentiation of decidual stromal cells, resulting in unordered regionalization of decidualization and compromised pregnancy success. The reduction of Bmp2 caused by aberrantly activated FGF2-ERK1/2 signaling pathway deranges the adequate development of the SDZ due to disorganized genomic H3K4me3 modification in the absence of Men1. PTX3, an extracellular protein trapping FGF2, is identified as a Menin directly regulated downstream gene and provides insightful information for the orchestration of decidualization and pregnancy maintenance in curbing the cross-talk between FGF2-ERK1/2 and BMP2.

## Results

**Men1 is expressed in the peri-implantation mouse uterus in a spatiotemporal manner**. Highly heterogeneous uterine endometrium is comprised of different cell types that undergo tremendous changes during early pregnancy. To elucidate the physiological significance of Men1 during early pregnancy, we first examined the cell-specific and temporal expression of Men1 in peri-implantation uteri using in situ hybridization. There was weak Men1 expression in the pre-implantation uterus on days 1 and 4. After embryo implantation, Men1 transcript was moderately upregulated in proliferating stromal cells surrounding the blastocyst on day 5. Accompanied by the initiation of decidualization, stromal cells underlying the implantation chamber exit mitosis and differentiate into epithelioid cells to form the PDZ. The transcript of Men1 became undetectable in the PDZ but was intensely expressed in proliferating stromal cells outside the PDZ on day 6, which was further supported by the co-staining of Ki67 and Menin (Fig. 1a and Supplementary Fig. 1a, b). After the SDZ was established by day 8, Men1 expression was restricted in the mesometrial (M) pole (Fig. 1a). These observations indicated that Men1 expression was tightly correlated with the progress of decidualization. Consistent with its mRNA expression pattern, Menin protein was widely expressed in the uterus during decidualization and located mainly in nucleus (Fig. 1b). It was interesting to note that Menin protein possessed wider expression than its transcript which is perhaps due to the discrepant stability between mRNA and protein (Fig. 1b). Account for the well overlapping between Men1 mRNA and proliferating cells, it was conceivable to speculate that the newly transcribed Men1 was probably for Men1 translation for daughter cells during mitosis. The spatiotemporal expression pattern of Men1 indicates that it could be a considerable regulator in peri-implantation events.

**Mice with uterine deletion of Men1 show severe subfertility**. To explore the functional role of uterine Men1 in pregnancy, we generated mice with uterine deletion of Men1(Men1$^{d/d}$) by crossing Men1 floxed mice (Men1$^{f/f}$) with PR-Cre mice (Pgr$^{Cre/+}$)[10,16]. These mice show efficient deletion of Men1 in the pregnant uterus at both mRNA and protein levels (Fig. 1c–e). To interrogate the role of Men1 in female fertility, Men1$^{d/d}$ mice and their littermate controls (Men1$^{f/f}$) were mated with wild type fertile males. Only 25% plug-positive Men1$^{d/d}$ mice gave birth to offspring and the litter size was significantly lower than those in littermate Men1$^{f/f}$ mice (Fig. 1f, g). These results demonstrated uterine Men1 was crucial for a successful pregnancy.

**Ablation of uterine Men1 causes defective decidualization and miscarriage at mid-gestation**. Early pregnancy is a highly programmed dynamic process involving blastocyst attachment, decidualization, and placentation. Defects at any stage will give rise to adverse pregnancy outcomes. To dissect out the defects causing subfertility in Men1$^{d/d}$ mice, we first assessed the early pregnancy events on day 5 with increased vascular permeability at the attachment site[17]. The number of implantation sites and the embryo spacing were comparable in Men1$^{f/f}$ and Men1$^{d/d}$ mice on day 5 as visualized by intravenous injection of Chicago blue (Fig. 2a, b). The gross morphology and embryo attachment reaction as revealed by H&E staining and COX2 expression, respectively, were also similar between Men1$^{f/f}$ and Men1$^{d/d}$ (Supplementary Fig. 2a and Fig. 2c). Stromal cells surrounding the blastocyst undergo extensive proliferation and differentiation into decidual cells under the initiation of BMP2[2,8]. The semblable proliferation and expression pattern of Bmp2 and BMP antagonist Dan in Men1$^{f/f}$ and Men1$^{d/d}$ implantation sites confirmed that mice with uterine deletion of Men1 had normal initiation of

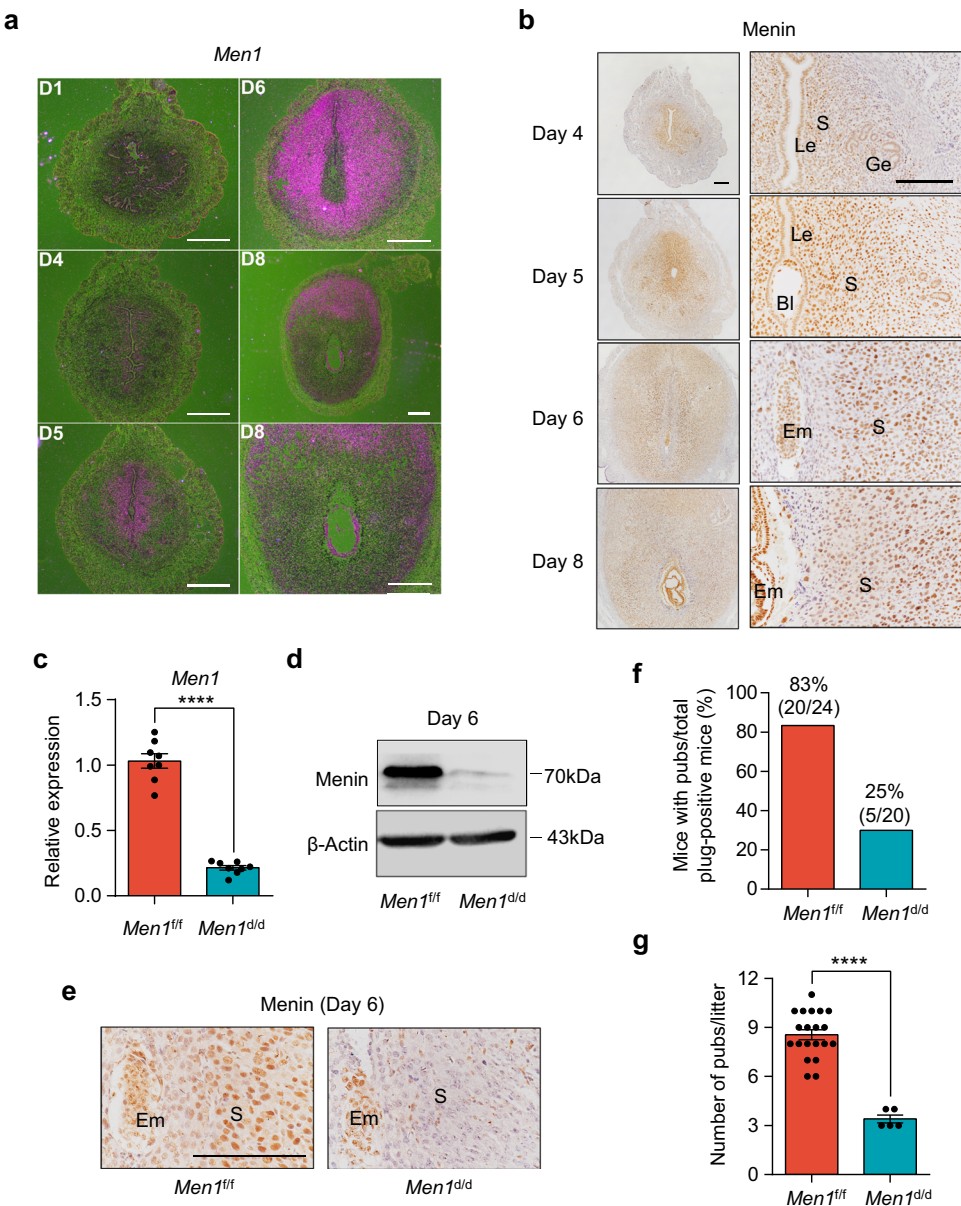

**Fig. 1 Uterine-specific deletion of *Men1* leads to female subfertility. a** In situ hybridization of *Men1* in WT uteri on days 1, 4, 5, 6, and 8 of pregnancy. Scale bars: 500 μm. **b** Immunohistochemical analysis of Menin in day 4 uteri and days 5, 6, and 8 implantation sites. Le luminal epithelium, Ge glandular epithelium, S stroma, Bl blastocyst, Em embryo. Scale bar: 200 μm. **c** Quantitative real-time PCR analysis of *Men1* mRNA levels in *Men1*^f/f^ and *Men1*^d/d^ implantation sites on day 6. The values are normalized to *Gapdh* and indicated as the mean ± SEM ($n = 8$ biologically independent samples). Two-tailed unpaired Student's *t*-test, ****$p = 4.26e-7$. **d** Immunoblotting analysis of Menin protein in *Men1*^f/f^ and *Men1*^d/d^ implantation sites on day 6. B-Actin was used as a loading control. **e** Immunohistochemical analysis of Menin in *Men1*^f/f^ and *Men1*^d/d^ implantation sites on day 6. S stroma, Em embryo. Scale bar: 200 μm. **f** Pregnancy rates in *Men1*^f/f^ and *Men1*^d/d^ female mice. The number within brackets indicates females with pups over total number of plug-positive females. **g** Average litter sizes in *Men1*^f/f^ ($n = 20$ animals) and *Men1*^d/d^ ($n = 5$ animals) mice. Data represent the mean ± SEM. Two-tailed unpaired Student's *t*-test, ****$p = 4.31e-8$.

decidualization (Fig. 2d and Supplementary Fig. 1b–e). Moreover, IGFBP5, which provides a favorable environment for blastocyst growth by coupling with IGF1[18], had a similar subepithelial expression pattern in *Men1*^f/f^ and *Men1*^d/d^ implantation sites (Supplementary Fig. 1f).

Since the deficiency of *Men1* has no severe defects in the blastocyst attachment reaction and the initiation of decidualization, we next examined the progress of decidualization. The morphology of the decidual bulge and the number of implantation sites were comparable between *Men1*^f/f^ and *Men1*^d/d^ mice on day 6 with the termination of stromal cell proliferation in the established PDZ (Fig. 2e, f and Supplementary Fig. 3a). Our

results showed that there were comparable expressions of OCT4, Ki67, Dtprp, COX2, *Bmp7*, β-Catenin, and *Wnt4* in day 6 uteri, implying normal embryo development, stromal cell proliferation, PDZ formation[19–22] (Supplementary Fig. 3b–h). We also noticed that the glands distribution was abnormal with increased number in the lateral areas near the M pole in stroma of *Men1*^d/d^ mice as characterized by FOXA2[23] from pregnant day 4–6 (Supplementary Fig. 4). Collectively, these results indicated that uterine *Men1* was dispensable for blastocyst attachment, implantation, and PDZ formation during early pregnancy.

Following the formation of PDZ, stromal cells adjacent to the PDZ continue to proliferate and differentiate to form SDZ until

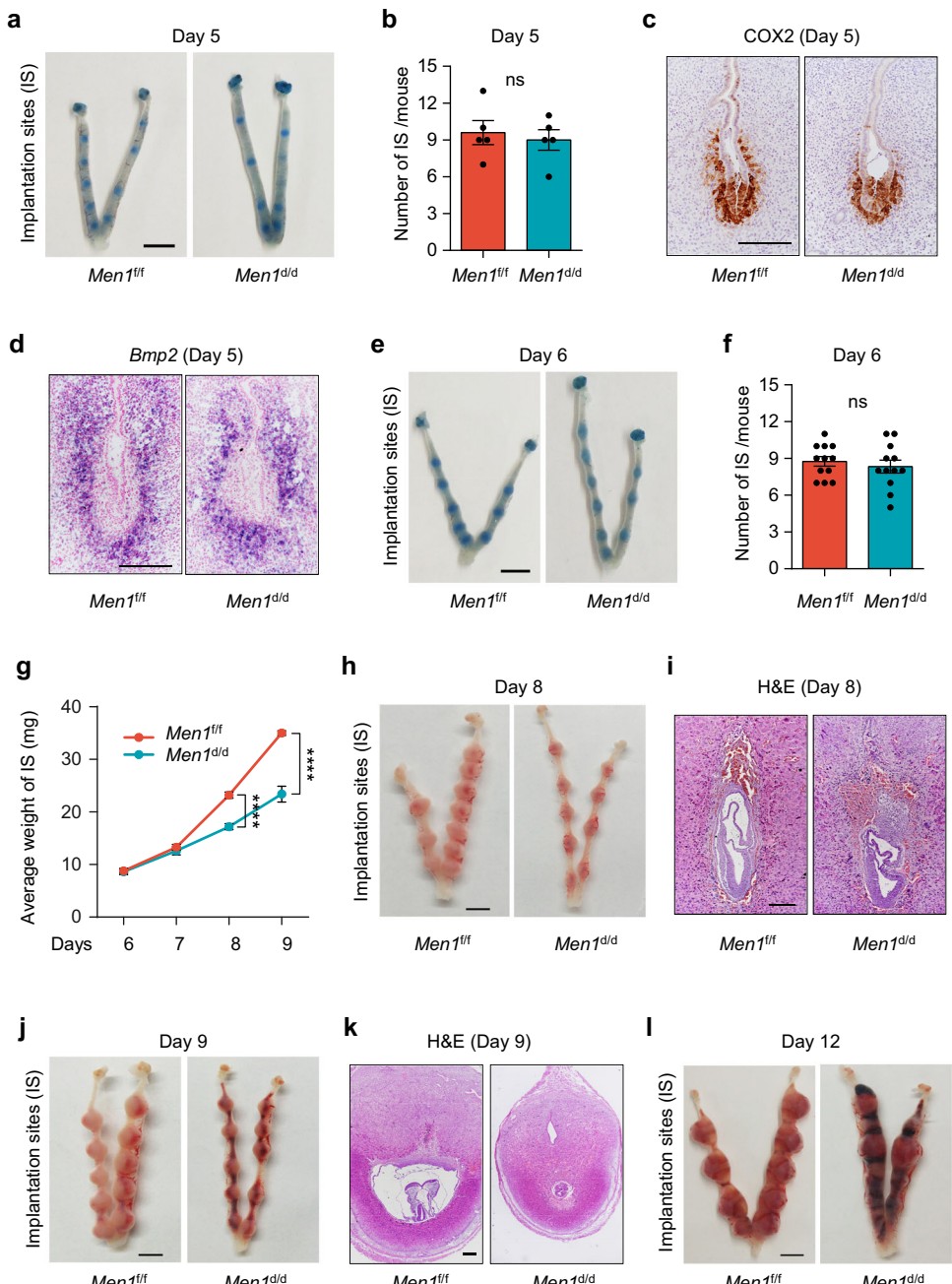

**Fig. 2 Mice with uterine *Men1* deletion show defective embryo development at the post-implantation stage. a** Implantation sites marked by Chicago blue dye solution in *Men1*^f/f and *Men1*^d/d mice on day 5. Scale bar: 5 mm. **b** The average number of implantation sites in *Men1*^f/f ($n = 5$ animals) and *Men1*^d/d ($n = 5$ animals) mice on day 5. Data represent the mean ± SEM. Two-tailed unpaired Student's *t*-test. ns, not significant. **c** Immunohistochemical analysis of embryo attachment reaction marker COX2 in *Men1*^f/f and *Men1*^d/d implantation sites on day 5. Scale bar: 200 μm. **d** In situ hybridization of *Bmp2* in *Men1*^f/f and *Men1*^d/d implantation sites on day 5. Scale bar: 200 μm. **e** Implantation sites marked by Chicago blue dye solution in *Men1*^f/f and *Men1*^d/d mice on day 6. Scale bar: 5 mm. **f** The average number of implantation sites in *Men1*^f/f ($n = 12$ animals) and *Men1*^d/d ($n = 12$ animals) mice on day 6. Data represent the mean ± SEM. Two-tailed unpaired Student's *t*-test. ns, not significant. **g** The average weight of the implantation sites in *Men1*^f/f ($n = 5$ animals) and *Men1*^d/d ($n = 5$ animals) mice on days 6, 7, 8, and 9 of pregnancy. Data represent the mean ± SEM. Two-tailed unpaired Student's *t*-test, ****$p = 8.59e\text{-}5$ (Day 8), ****$p = 7.65e\text{-}5$ (Day 9). **h** Representative images of day 8 pregnant uteri in *Men1*^f/f and *Men1*^d/d mice. Scale bar: 5 mm. **i** Histology of embryo development in *Men1*^f/f and *Men1*^d/d implantation sites on day 8. Scale bar: 200 μm. **j** Representative images of day 9 pregnant uteri in *Men1*^f/f and *Men1*^d/d mice. Scale bar: 5 mm. **k** Histology of implantation sites in *Men1*^f/f and *Men1*^d/d mice on day 9. Scale bar: 200 μm. **l** Representative images of day 12 pregnant uteri in *Men1*^f/f and *Men1*^d/d mice. Scale bar: 5 mm.

day 8. The reduced average weight of implantation sites with retarded embryos was emerged in *Men1*^d/d mice on day 8 in comparison to *Men1*^f/f mice (Fig. 2g–i). From day 9–12, more obvious defects appeared at the implantation sites, including abnormal placentation and severe embryo absorption (Fig. 2j–l). These data hinted that the development of SDZ was hindered from day 7–8. In order to further confirm the effect of *Men1* deletion in the progress of decidualization, an oil-induced

artificial decidualization model[24] was applied (Supplementary Fig. 5a). Men1[f/f] mice uteri exhibited a robust decidual response 96 h after intraluminal oil injection, while the Men1[d/d] uteri showed a remarkably weakened or disappeared decidual response (Supplementary Fig. 5b, c). Collectively, these results demonstrated that uterine-specific deletion of Men1 disturbed the progression of decidualization and ultimately resulted in embryo resorption and miscarriage at mid-gestation.

**Uterine deletion of Men1 leads to inadequate differentiation of decidual cells in the SDZ.** The observations of abnormal decidualization on day 8 led us to investigate the cellular defects in Men1[d/d] uteri. Polyploidization, a hallmark of mature decidual cells in the SDZ, was marked by mono or bi-nucleated cells through endoduplication. Indeed, the ablation of Men1 in uteri resulted in decreased decidual cell polyploidy and smaller nuclear size (Fig. 3a). H3K27me3 was critical epigenetic modification for X chromosome inactivation for dose complementary effect of female somatic cells[25]. Our results showed that polyploid decidual cells in Men1[f/f] mice possessed >4 puncta H3K27me3 staining, indicating at least four sets of homologous chromosomes in polyploid decidual cells (Fig. 3b). While the number of polyploid decidual cells with four or more puncta H3K27me3 staining was obviously reduced in Men1[d/d] mice (Fig. 3b). Consistently, we found aberrantly increased PCNA (DNA replication in S-phase) and BrdU (DNA synthesis in S-phase) in Men1[d/d] decidual cells on day 8 of pregnancy (Fig. 3b). Further, flow cytometric analysis of DNA content revealed that 2 N cells were significantly increased with a concomitant decrease of >4 N cells in Men1 deletion decidual tissues on day 8 (Fig. 3c, d and Supplementary Fig. 6). In Men1[d/d] mice, both Dtprp mRNA and protein were significantly reduced on day 8 (Fig. 3e, f), suggesting that the terminal differentiation of decidual cells was impaired. These results indicated Men1 was critical for the progression of polyploidy during decidualization.

On day 8, the expression of COX2 in the presumptive site of placentation was disordered, foreboding impaired placentation in Men1[d/d] mice (Fig. 3g, h). Accompanied by normal decidualization, abundant uterine natural killer (uNK) cells accumulated at the mesometrial pole on day 8[26]. In contrast to numerous DBA+ uNK cells in Men1[f/f] mesometrial pole, Men1 null uteri exhibited dramatic decrease of uNK cells (Fig. 3h). Aberrant decidualization tends to give rise to abnormal trophoblast invasion and compromised placentation[5]. Eccentric expression pattern of trophoblast giant cells marker PL1 in Men1[d/d] uteri signified shallow trophoblast invasion on day 8 (Fig. 3i). All these observations strongly proved the defective decidualization in the absence of uterine Men1.

The morphological and physiological changes of stromal cells during decidualization are mainly regulated by steroid hormones from ovary[1]. Progesterone is indispensable for decidualization maintenance through its cognate nuclear receptor progesterone receptor (PR)[5]. There was no significant difference in circulating levels of E2 and P4 in both groups on day 6 with similar expression of progesterone biosynthetic enzymes cytochrome P450 cholesterol side-chain cleavage enzyme (P450scc) and 3β-hydroxysteroid dehydrogenase (3β-HSD) in the ovary as well as comparable expression PR and its target genes, Hand2 and Hoxa10, in day 6 and day 8 uterus (Supplementary Fig. 7a, b and Supplementary Fig. 8a–f). These data suggested that the defect decidualization was primarily originated from local dysfunction of Men1 in stromal cells.

**Procedures for decidualization dysregulated in the absence of uterine Men1.** To further excavate the underlying molecular

mechanism for the defective terminal differentiation of decidual cells in the SDZ observed in Men1[d/d] mice, decidual tissues were isolated from Men1[f/f] and Men1[d/d] uteri on day 8 and subjected to gene expression profiles analysis using RNA sequencing (RNA-seq). A total of 1669 genes were differentially expressed (RPKM > 0.5, p < 0.05) with 870 downregulated genes (Supplementary Data 1) and 789 up-regulated genes (Supplementary Data 2) in Men1[d/d] mice in comparison to wild type (Fig. 4a). Decidualization involves a series of biological changes including the proliferation and differentiation of stromal cells, angiogenesis, and immunological adaptation[5,22,27]. Functional annotation revealed that downregulated genes upon Men1 deletion mainly enriched in the terms of cell proliferation inhibition, fatty acid metabolic process, multicellular organism development, and negative regulation of angiogenesis (Fig. 4b and Supplementary Data 3). In contrast, genes up-regulated in Men1[d/d] decidual tissues were enriched in categories that were resistant to cell differentiation such as positive regulation of ERK1/2 cascade and categories detrimental to the development of decidua such as apoptotic process and inflammatory response (Fig. 4b and Supplementary Data 4).

The downregulated genes in negative regulation of cell proliferation, including Bmp2, Ptn, Fgf10, and Igf1, possessed specific expression patterns and were important for decidualization[6,28,29] (Fig. 4c, d). On the contrary, genes associated with the activation of MAPK activity that promotes cell proliferation (Fgfr1, Pdgfa) were upregulated in Men1[d/d] uteri (Fig. 4c). Besides, genes expressed in the thin layer of undecidualized stromal cells were increased after Men1 knockouts, such as Sfrp4 and Prlr (Fig. 4c, d).

Increased vascular permeability and angiogenesis are crucial to successful decidualization. Angiopoietins (ANGPT1 and ANGPT2) and their receptor TIE2 directs angiogenesis during decidualization[22,27]. ANGPT1 induces vessel maturation and maintains vessel leakiness, whereas ANGPT2, acting as a natural antagonist of ANGPT1, induces vessel destabilization required for further sprouting and regression of vessels[30]. Our RNA-seq data indicated several genes in promoting angiogenesis (Angpt1, Edn1, Ednra) were upregulated, and genes in inducing destabilization of blood vessels and endothelial sprouting (Angpt2, Hif3a, Mmp2) were downregulated due to uterine Men1 deletion (Fig. 4c, e). Furthermore, NK-specific genes (Prf1, Klrg1, Ctsg, Il15r) and several granzyme genes (Gzma, Gzmb, Gzmc, Gzmd, Gzme, Gzmg) were significantly decreased in Men1[d/d] decidua (Fig. 4c), indicating the maturation of uNK was compromised. Moreover, Corin, functions in promoting trophoblast invasion and spiral artery remodeling[31], was decreased substantially in Men1[d/d] females (Fig. 4c).

Interestingly, loss of Men1 led to decreased Cdkn1b (encoding p27) in both mRNA and protein levels (Fig. 4c, f). Especially, the immunohistochemical analysis indicated that the expression of p27 was higher in polyploid cells (Fig. 4f). Previous studies manifest that increased p27 level in liver cells causes a failure to enter mitosis and thereby induces polyploidy[32]. Reduced polyploid decidual cells in Men1[d/d] decidual tissues might be accountable for the reduction of p27. In addition, gene set enrichment analysis (GSEA) analysis showed that cytokinesis-associated genes (Rhoc, Rhob, Dstn)[33] were significantly enriched by Men1 loss (Fig. 4g, h and Supplementary Data 5). Consistently, there were more stromal cells undergoing proliferating in Men1 knockout SDZ as evidenced by significant increase of mitotic marker pH3 (Fig. 4i, j). Above all, Men1 loss had a strong negative impact on normal physiological changes in decidualization.

**Men1-deficient uteri show diminished BMP responsiveness but enhanced FGF responsiveness.** As aforementioned, the uterine

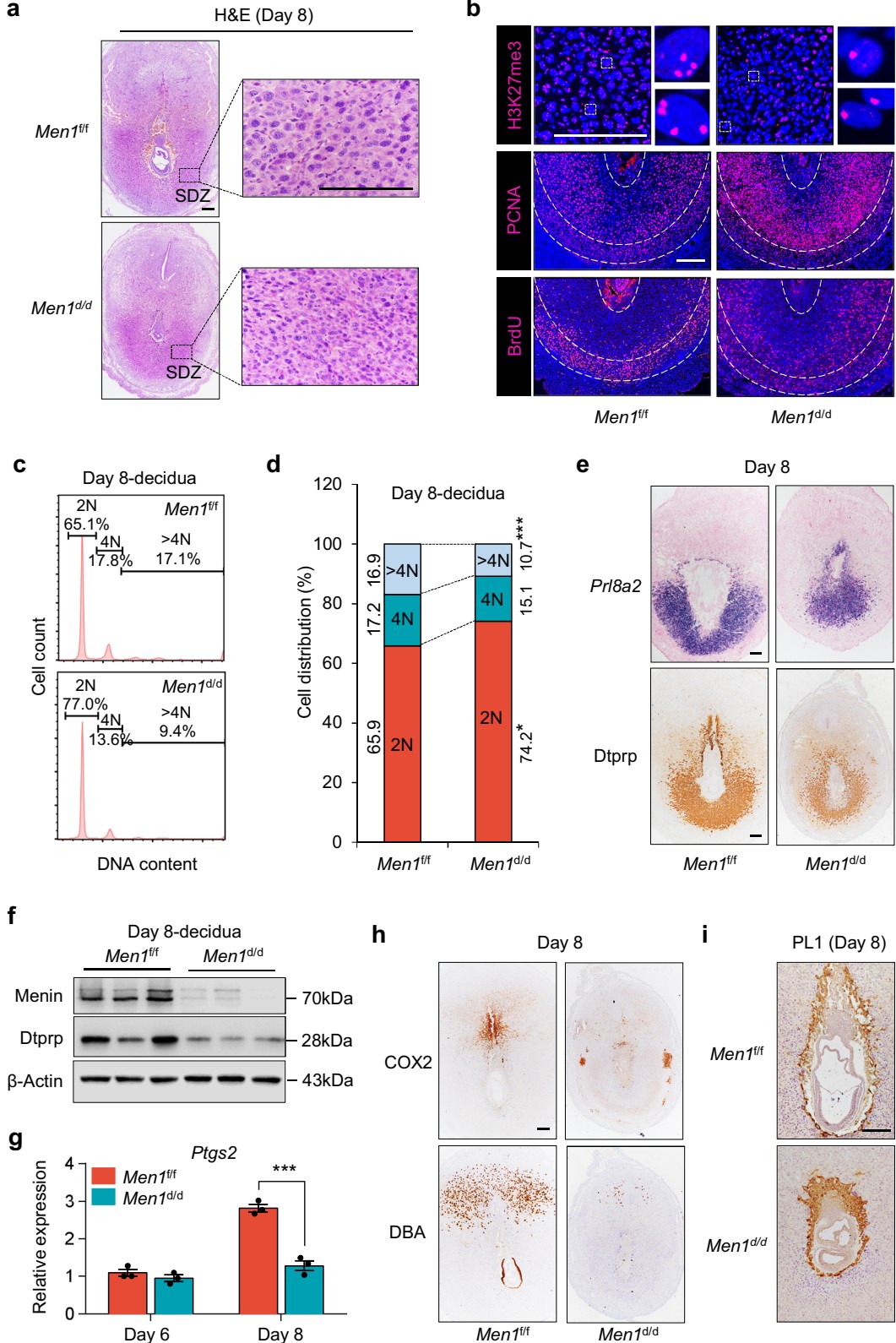

with *Men1* deletion had an insufficient terminal differentiation of stromal cells in the SDZ. Interestingly, the FGF signaling pathway that positively regulated cell proliferation was significantly enriched among the upregulated genes in *Men1*^d/d uteri. In contrast, the gene set responding to BMP which motivated differentiation

of stromal cells was significantly suppressed in *Men1*^d/d uteri (Fig. 5a, b and Supplementary Data 5). During decidualization, *the Fgf2* transcript was localized in the undifferentiated stromal cells underneath the myometrium[6]. FGF2 binds and activates the tyrosine kinase receptor complex FGFR1-FRS2 and recruits the

**Fig. 3 Uterine *Men1* deletion compromises the terminal differentiation of stromal cells during decidualization. a** Histology of day 8 implantation sites and polyploid stromal cells in *Men1*[f/f] and *Men1*[d/d] mice. Scale bars: 200 μm. SDZ, secondary decidual zone. **b** Immunofluorescence analysis of H3K27me3, PCNA, and BrdU in the SDZ of *Men1*[f/f] and *Men1*[d/d] mice on day 8. Scale bars: 200 μm. **c** DNA content quantification of decidual cells by FACS from *Men1*[f/f] and *Men1*[d/d] mice on day 8. **d** Cellular distribution for 2 N, 4 N, and >4 N populations of stromal cells in *Men1*[f/f] (*n* = 3 animals) and *Men1*[d/d] (*n* = 3 animals) implantation sites on day 8. Data represent the mean ± SEM. Two-tailed unpaired Student's *t*-test, \**p* = 0.021 (2 N), \*\*\**p* = 9.97e-4 (>4 N). **e** The expression of decidual marker Dtprp in *Men1*[f/f] and *Men1*[d/d] implantation sites on day 8. In situ hybridization indicates the expression of *Prl8a2* mRNA (top panel). Immunohistochemical staining indicates the expression of protein Dtprp (bottom panel). Scale bar: 200 μm. **f** Immunoblotting analysis of Dtprp in *Men1*[f/f] and *Men1*[d/d] implantation sites on day 8. B-Actin was used as a loading control. **g** Quantitative real-time PCR analysis of *Ptgs2* in *Men1*[f/f] and *Men1*[d/d] implantation sites on day 6 and day 8. The values are normalized to *Gapdh* and indicated as the mean ± SEM (*n* = 3 biologically independent samples). Two-tailed unpaired Student's *t*-test, \*\*\**p* = 1.82e-4. **h** Immunohistochemical analysis of COX2 and DBA staining in *Men1*[f/f] and *Men1*[d/d] implantation sites on day 8. Scale bar: 200 μm. **i** Immunohistochemical analysis of PL1 in *Men1*[f/f] and *Men1*[d/d] implantation sites on day 8. Scale bar: 200 μm.

downstream GRB2-SOS1 complex to incite ERK1/2 signaling cascade which is a major pathway controlling cell proliferation[34]. *Fgfr1* and *Frs2* but not the ligand *Fgf2* were significantly increased by *Men1* deletion (Fig. 5b, c). BMP2 is a well-known decidual marker critical for the differentiation of stromal cells during decidualization[6,8]. The expression of *Bmp2* and its target *Id1* was substantially lower in *Men1*[d/d] mice uteri on day 8 of pregnancy (Fig. 5b, d). BMPs execute their effects by inducing the phosphorylation of SMAD1/5/8 proteins. Accordingly, the protein level of p-SMAD1/5/8 was decreased in *Men1*[d/d] decidual cells (Fig. 5e and Supplementary Fig. 9). Interestingly, both the significant elevated expression of *Fgfr1* and reduced expression of *Bmp2* caused by *Men1* deletion mainly occurred in the SDZ (Fig. 5f). We also observed an elevation of phosphorylated ERK1/2 in *Men1*[d/d] stromal cells in the SDZ (Fig. 5g). These observations were consistent with enrichment of ERK1/2 cascade and decreased cell differentiation observed in *Men1*[d/d] decidual tissues (Fig. 4b, c). Thus, spatiotemporal decidualization regulated by programmed FGF signaling and BMP signaling was disorganized in *Men1* deficiency uterine. Indeed, mutually inhibitory effects between FGF and BMP signaling are common modules throughout embryogenesis[35]. Next, we explored whether there was a cross-talk between BMP2 and FGF2-ERK1/2 signals, and what role Menin played in this communication during the development of the SDZ.

**Menin positively regulates the expression of *Ptx3* in an H3K4me3-dependent manner.** Menin is an indispensable component of MLL1/2 HMT complex that catalyzes H3K4me3 in a locus-specific manner, which is usually associated with transcriptional activation of genes[9,36]. We speculated that Menin activated gene expression through H3K4me3 during decidualization. Total H3K4me3 protein level was not affected by *Men1* ablation as previously reported[14,37,38] (Supplementary Fig. 10a). Although deficiency of *Men1* does not disrupt global H3K4me3, Menin regulates the H3K4me3 levels across regions proximal to TSSs of a limited number of cell-specific genes[38,39]. In order to define its relevance to specific gene expression in decidual cells, we performed ChIP-seq for Menin and H3K4me3 in decidual tissues on day 8. Consistent with the published literature, Menin and H3K4me3 is predominantly positioned at gene promoter (Menin peaks: *n* = 13,347, 76.65% at promoters; H3K4me3 peaks: *n* = 29,040, 70.62% at promoters) (Fig. 6a). Almost 80% of peaks bound by Menin were also H3K4me3 modified (Supplementary Fig. 10b). Combined with RNA-seq data, we found that Menin and H3K4me3 were significantly enriched at the transcription start site (TSS) of expressed genes (RPKM > 1) (Fig. 6b and Supplementary Fig. 10c). A direct comparison of Menin and H3K4me3 ChIP-seq peaks at TSSs showed a strong positive correlation ($R^2$ = 0.5888291) (Fig. 6c).

Specifically, read coverage of H3K4me3 around TSSs was modestly reduced in the absence of *Men1* (Fig. 6b, d). These results suggested that Menin is involved in H3K4me3 modification across regions proximal to TSSs during decidualization. We, therefore, focused on the potential function of Menin-dependent H3K4me3 in transcriptional activation of specific genes during decidualization.

Analysis of ChIP-seq data revealed that although Menin-H3K4me3 typical target genes identified in other cell types (e.g. *Cdkn1b*, *Cdkn2c*, *Esr1*, *Hoxa* cluster, and *Meis1*) were also bound by Menin in decidual tissue, none of them had H3K4me3 reduction after *Men1* deletion at the promoters (Fig. 6e). Correspondingly, except for the decreased expression of *Cdkn1b*, the mRNA levels of other Menin target genes had no significant difference in the absence of *Men1* (Figs. 4c, 6e). There might be an alternative mechanism different from the H3K4me3 dependent mechanism as previously reported participated in the transcriptional regulation of *Cdkn1b* by Menin[13]. Researches investigating genome-wide function of Menin yielded cell-specific results in terms of the expression of H3K4me3 dependent target genes[15,38–40]. According to these results, we hypothesized that *Men1* regulates the transcription of a subset of decidual-specific target genes through H3K4me3 modification. H3K4me3 peaks at regions proximal to TSSs were classified into two clusters depending on the changes after *Men1* deletion (Fig. 6f and Supplementary Fig. 10d). Cluster 1 contains loci where H3K4me3 ChIP-seq signal was decreased following *Men1* deletion, representing 6469 associated genes (Fig. 6f). While loci in cluster 2 have no significant change of H3K4me3 levels, representing 5644 associated genes (Supplementary Fig. 10d). To identify the direct Menin-H3K4me3 target genes responsible for the decidualization defects observed in *Men1*[d/d] uteri, we intersected the RNA-seq data with genes bound by Menin as well as 6469 genes in cluster 1 and found 2995 potential Menin-H3K4me3 direct target genes. Among the 348 genes that were differentially expressed after *Men1* deletion, only 217 genes had significantly reduced mRNA levels (Fig. 6g). We listed the top-ranked downregulated genes that are most likely to be the targets of Menin-H3K4me3 in decidual cells (Fig. 6h, i). We also determined the H3K4me3 accumulation in promoters of Menin target genes (e.g. Sfrp2, Twist2, Snx10, and 378 Adora2b) by ChIP-qPCR assays, and it displayed a reduced H3K4me3 modification in decidual cells upon uterine *Men1* deletion (Supplementary Fig. 11a–d). These results suggested that loss of *Men1* resulted in reduction of H3K4me3 at regions proximal to TSSs of a very limited number of genes expressed in decidua cells.

In these potential target genes, *Ptx3* attracted our attention since the deletion of *Ptx3* resulted in compromised decidualization[41] (Fig. 6g). Indeed, the peak of H3K4me3 was significantly reduced at the *Ptx3* locus, and RNA-seq tracks for

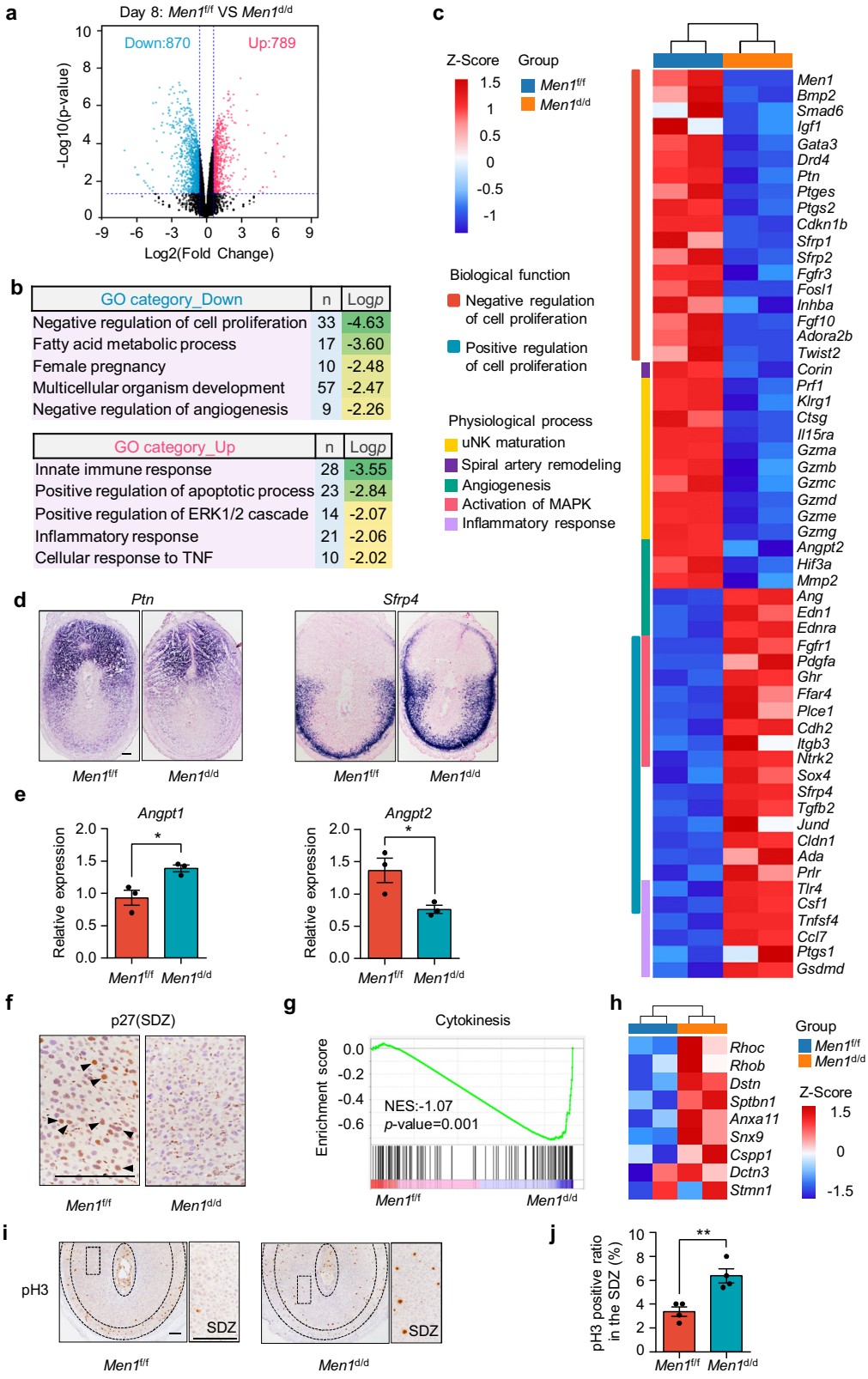

**a** Day 8: *Men1*^f/f VS *Men1*^d/d

**b**

| GO category_Down | n | Log*p* |
|---|---|---|
| Negative regulation of cell proliferation | 33 | -4.63 |
| Fatty acid metabolic process | 17 | -3.60 |
| Female pregnancy | 10 | -2.48 |
| Multicellular organism development | 57 | -2.47 |
| Negative regulation of angiogenesis | 9 | -2.26 |

| GO category_Up | n | Log*p* |
|---|---|---|
| Innate immune response | 28 | -3.55 |
| Positive regulation of apoptotic process | 23 | -2.84 |
| Positive regulation of ERK1/2 cascade | 14 | -2.07 |
| Inflammatory response | 21 | -2.06 |
| Cellular response to TNF | 10 | -2.02 |

*Ptx3* were profoundly decreased in *Men1*^d/d uteri following *Men1* deletion (Fig. 7a). *Men1* loss decreased *Ptx3* expression in decidual tissues on day 8 of pregnancy (Fig. 7b). Through Menin ChIP assays using a pair of specific primers targeting the *Ptx3* promoter locus, we further confirmed that *Ptx3* was a direct target gene of Menin in decidual cells (Fig. 7c). Meanwhile, H3K4me3 occupancy at the *Ptx3* promoter locus was dramatically decreased in *Men1*-knockout stromal cells (Fig. 7d). These results suggest that Menin upregulates *Ptx3* expression in an H3K4me3 dependent manner.

**Fig. 4 Procedures for decidualization deregulated by uterine _Men1_ loss. a** Volcano plot of differentially expressed genes in _Men1_[f/f] and _Men1_[d/d] decidual tissues on day 8 as determined by RNA-seq. Blue and red dots represent genes that are differentially expressed ($p < 0.05$) between _Men1_[f/f] and _Men1_[d/d] decidual tissues by exact negative binomial test in edgeR package of R. Genes not significantly differentially expressed are shown in black. **b** Gene Ontology functional analysis for differentially expressed genes determined by DAVID by Kappa Statistics ($p < 0.05$). n number of genes. **c** Heatmap of physiological function associated genes differentially regulated between _Men1_[f/f] and _Men1_[d/d] uteri on day 8. RPKM of each gene from different samples was normalized to Z-score. **d** In situ hybridization of _Ptn_ and _Sfrp4_ in _Men1_[f/f] and _Men1_[d/d] implantation sites on day 8. Scale bar: 200 μm. **e** Quantitative real-time PCR analysis of _Angpt1_ and _Angpt2_ mRNA levels in _Men1_[f/f] and _Men1_[d/d] implantation sites on day 8. The values are normalized to _Gapdh_ and indicated as the mean ± SEM ($n = 3$ biologically independent samples). Two-tailed unpaired Student's _t_-test, *$p = 0.025$ (_Angpt1_), *$p = 0.039$ (_Angpt2_). **f** Immunohistochemical analysis of p27 in the SDZ of _Men1_[f/f] and _Men1_[d/d] implantation sites on day 8. Scale bar: 200 μm. **g** GSEA plot showing the enrichment of cytokinesis-related genes ($n = 115$) in _Men1_[d/d] uteri compared with _Men1_[f/f] uteri on day 8. NES normalized enrichment score. ***$p = 0.001$. **h** Heatmap of cytokinesis associated genes differentially expressed between _Men1_[f/f] and _Men1_[d/d] mice uteri on day 8. RPKM of each gene from different samples was normalized to Z-score. **i** Immunohistochemical analysis of pH3 in _Men1_[f/f] and _Men1_[d/d] implantation sites on day 8. Scale bars: 200 μm. **j** Quantitative analyses of pH3 positively stained cells in the SDZ on day 8. Data represent the mean ± SEM ($n = 4$ biologically independent samples). Two-tailed unpaired Student's _t_-test, **$p = 0.0051$.

**Menin inhibits the cross-talk between FGF2-ERK1/2 and BMP2 by regulating the transcription of _Ptx3_.** PTX3 has been shown to bind with selected FGFs, including FGF2 and FGF8b, through its N-terminal domain, and sequester them in the extracellular matrix, thus inhibiting their biological effects[42]. As described above, _Men1_ loss upregulated FGF response. We found _Ptx3_ specifically expressed in the SDZ close to the thin layer of undifferentiated stromal cells where _Fgf2_ located and significantly decreased following _Men1_ deletion (Fig. 7e). During embryogenesis, FGF2 is considered to be a paracrine factor and is essential for tissue patterning and organogenesis[35]. Based on these results, we considered that the decrease of _Ptx3_ caused by _Men1_ deficiency contributed to excessive ERK1/2 activation induced by paracrine FGF2 produced by the layer of undifferentiated stromal cells. To confirm this conjecture, primary mouse endometrial stromal cells (mESC) isolated from day 4 uterus of pseudopregnancy were cultured with or without recombinant mouse PTX3 in the presence of FGF2. PTX3 treatment for 72 h significantly inhibited the phosphorylation of ERK1/2 induced by FGF2 in mESC compared with cells without PTX3 treatment (Fig. 7f). Consistent with previous studies, the inhibitory effect of PTX3 on FGF2 was conserved during mouse decidualization. Thus, we concluded Menin suppressed the activation of FGF2-ERK1/2 signaling pathway in the SDZ by modulating _Ptx3_ transcription through epigenetic H3K4me3 modification.

In the preceding results, _Men1_ loss decreased BMP response program (Fig. 5). Indeed, suppression of BMP signaling by FGF2 sustained undifferentiated proliferation of human ES cells[43]. Next, we explored whether the reduction of BMP2 was caused by abnormal highly activated FGF2-ERK1/2 signal in the absence of _Men1_. During in vitro decidualization of mESC induced by E2 and P4, the expression pattern of Menin was almost consistent with that of BMP2 in spite of the slight decrease expression of Menin after 96 h of decidualization (Supplementary Fig. 12a, b). MI-503, a Menin inhibitor, blocks the Menin/MLL interaction and triggers Menin protein degradation via ubiquitin-proteasome pathway[44]. MI-503 treatment for 48 h reduced the expression of Menin protein without affecting the mRNA level of _Men1_ during the decidualization of mESC (Supplementary Fig. 12c, d). The expression of Menin target gene _Ptx3_ was also significantly reduced upon MI-503 treatment (Fig. 7g). However, MI-503 treatment for 72 h did not affect the expression of BMP2 and the activity of ERK1/2 during mESC decidualization (Supplementary Fig. 12e), implying _Bmp2_ was not a direct target of Menin and consistent with the ChIP data showing no Menin occupying and comparable H3K4me3 modification at _Bmp2_ promoter upon _Men1_ knockout (Supplementary Fig. 12f). While in the presence of exogenous FGF2, MI-503 treatment resulted in decreased BMP2 and elevated ERK1/2 phosphorylation (Fig. 7h), highlighting the significance of FGF2

for regulating BMP2 expression and appropriate decidualization. Meanwhile, co-treatment with MEK inhibitor PD0325901 reversed the reduction of BMP2 induced by MI-503 (Fig. 7h). These results demonstrated FGF2 acting as a paracrine factor had an inhibitory effect on BMP2 expression during decidualization. Supplementation of exogenous PTX3 also inhibited excessive activation of ERK1/2 and recovered BMP2 level (Fig. 7h). These results suggested Menin regulated the expression of BMP2 during decidualization by setting PTX3 as a trap for the FGF2-ERK1/2 pathway, thereby ensuring the success of decidualization (Fig. 8a, b).

## Discussion

The regulatory machinery of endometrial heterogeneity and decidua regionalization after embryo implantation has disturbed scientists for a long time due to the scarcity of a suitable mouse model. Developmentally associated transcription factors HOXA10, whose expression expanded throughout the PDZ and SDZ, have been reported to play an important role in establishing region-specific decidual tissue during decidualization[7,45]. While, other potential participants in decidua regionalization remained largely unknown. Here, we demonstrated that uterine _Men1_, which was spatiotemporally expressed during decidualization, guided the stromal cells towards the fate of terminal differentiation by directing histone modification rewiring, guaranteeing the expansion of the SDZ boundary during regionalization of decidualization.

Previous studies indicated that spatiotemporal expression pattern of _Bmp2_ with the progression of decidualization contributed to highly heterogeneous decidual cells[1,6]. At the initiation of decidualization, distinct accumulation of _Bmp2_ in the subepithelial stroma was essential for blastocyst attachment reaction at the implantation site[2,6,8]. With the spatiotemporal progression of decidualization, expanded _Bmp2_ expression in the PDZ and SDZ was critical for the terminal differentiation of stromal cells[6,8]. A significant decrease of _Bmp2_ in the SDZ on day 8 caused by _Men1_ deficiency disturbed the terminal differentiation of stromal cells during the expansion of SDZ without affecting the blastocyst attachment reaction and the PDZ formation.

Menin has yielded cell-specific results in terms of H3K4me3 regulation and the expression of target genes including _Hoxa_ genes, _Cdkn1b_ and _Cdkn2c_, _Esr1_, and _p35_ through connecting with histone methyltransferases MLL1/2[11-15]. There was a dramatic whole genomic chromatin alteration in human endometrium decidualization marked by histone modification and chromatin accessibility[46,47]. However, the role of H3K4me3 modification in decidual regionalization remained elusive. Distinctive histone methyltransferases in large COMPASS-like complexes performed non-redundant histone modifications in various cell types[48]. The SET1A/B family is responsible for genome-wide bulk H3K4me2/H3K4me3 modification and the

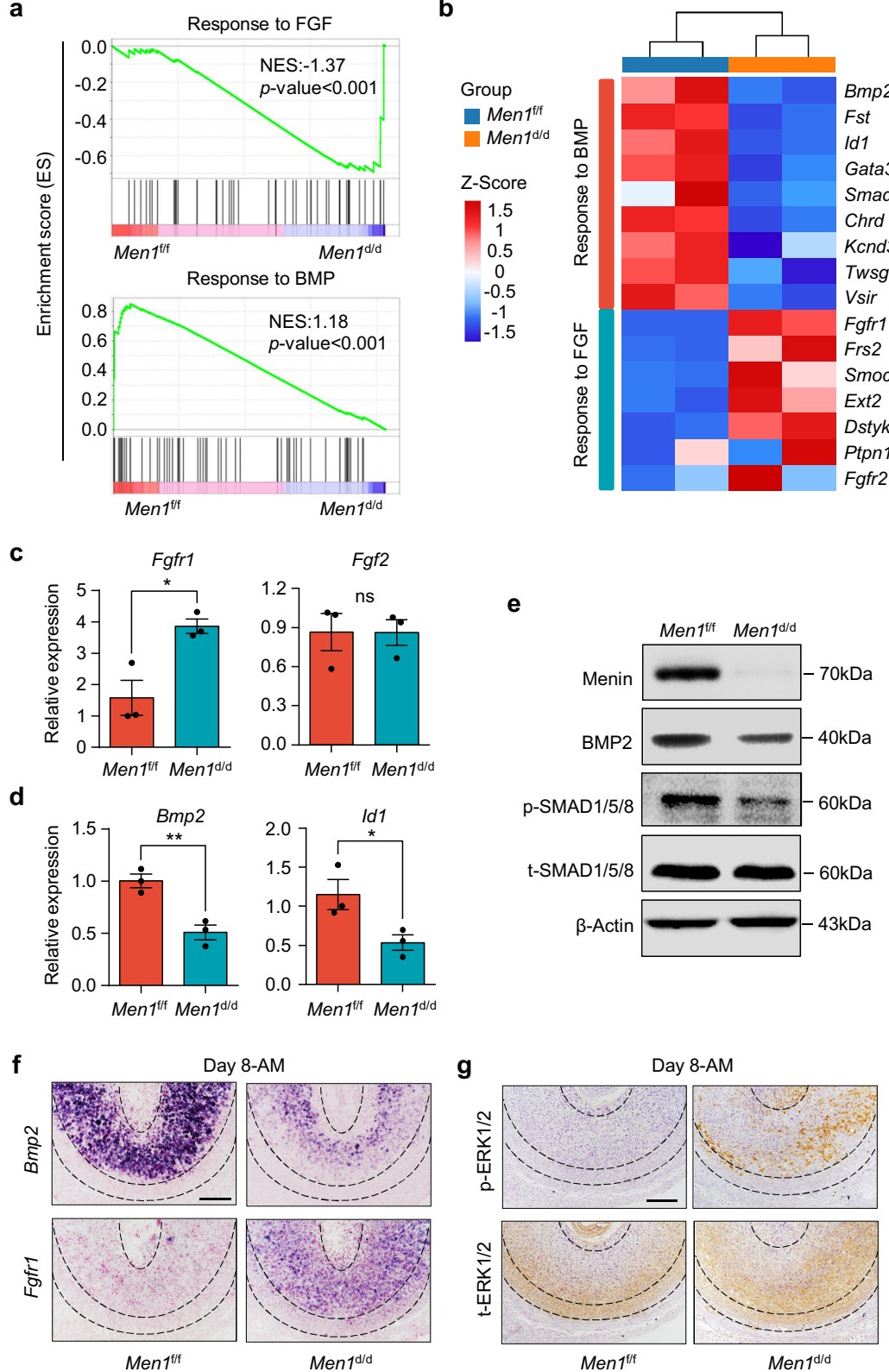

MLL3/4 complex family mediates H3K4me1 modification at enhancers and super-enhancer[48]. MLL1/2 complex, containing unique subunit Menin, is responsible for H3K4me3 modification at certain genes in cell-specific manner[11,48]. In this study, we depicted the H3K4me3 profiling in regionalized decidualization during early pregnancy and unraveled its regulatory apparatus by Menin.

Among the direct targets marked by Menin-H3K4me3 during stroma decidualization, *Ptx3* involved in decidualization regulation possesses specific expression pattern in the SDZ[41], but its physiological significance in stromal cell differentiation remains intangible. Several pieces of evidence showed that PTX3 acted as a natural FGF2 trap through its N-terminal domain to sequester FGF2-ERK1/2 signaling from promoting cell proliferation[42].

**Fig. 5 Decidual cells in *Men1*-deficient uteri show diminished BMP responsiveness and enhanced Fgf2 responsiveness. a** GSEA plot evaluating the changes in the indicated gene signatures of response to FGF (n = 46) and response to BMP (n = 57) in *Men1*^f/f uteri compared with *Men1*^d/d uteri. NES, normalized enrichment score. **b** Heatmap of differentially expressed genes between *Men1*^f/f and *Men1*^d/d uteri on day 8. RPKM of each gene from different samples was normalized to Z-score. **c**, **d** Quantitative real-time PCR analysis of *Bmp2*, *Id2*, *Fgfr1*, and *Fgf2* mRNA levels in *Men1*^f/f and *Men1*^d/d uteri on day 8. The values are normalized to *Gapdh* and indicated as the mean ± SEM (n = 3 biologically independent samples). Two-tailed unpaired Student's t-test, *p = 0.02 (*Fgfr1*), **p = 0.007 (*Bmp2*), *p = 0.046 (*Id2*). ns not significant. **e** Immunoblotting analysis of BMP2 and p-SMAD1/5/8 in *Men1*^f/f and *Men1*^d/d uteri. B-Actin and t-SMAD1/5/8 were used as loading controls. **f** In situ hybridization of *Bmp2* and *Fgfr1* in *Men1*^f/f and *Men1*^d/d uteri on day 8. AM antimesometrial pole. Scale bar: 200 μm. **g** Immunohistochemical staining of phosphorylated ERK1/2 and total ERK1/2 in *Men1*^f/f and *Men1*^d/d uteri on day 8. Scale bar: 200 μm.

Interestingly, our investigation demonstrated that *Men1*-deficient uteri showed augmented FGF2-FGFR1-ERK1/2 signaling activity in the SDZ in the absence of PTX3. This abnormality subsequently reduced *Bmp2* and blocked cell differentiation in the SDZ. Consistent with previous studies, the inhibitory effect of PTX3 on FGF2 was conserved during mouse decidualization. Indeed, the homeostasis between BMP and FGF has been described as "constituting a common module" throughout embryogenesis[35,49]. FGF signaling inhibited the expression of BMPs in dorsal mesoderm, thus limiting their expression to ventral mesoderm for proper dorsoventral axis specification[50]. Inhibition of BMP activity by the FGF signal promoted posterior neural development[51]. Furthermore, suppression of BMP signaling by FGF2 sustained undifferentiated proliferation of human ES cells[43]. Consistently, we evidenced that the discouraged cross-talk between BMP2 and FGF2 coordinated by Menin regulated PTX3 facilitated the maturation of decidua, which was further supported by the mutually exclusive expression of *Bmp2* and *Fgf2* in the SDZ and undifferentiated stromal cells, respectively. Together, we provided insightful evidence that the uncomfortable cross-talk between stromal cells in different regions through paracrine manner was curbed under the precise orchestrating of PTX3.

Moreover, *Men1* deficiency disrupted the polyploidization in the SDZ with reduced large mono or bi-nucleated polyploid cells as evidenced by H3K27me3. In female somatic cells, the punctate immunostaining of H3K27me3 stands for inactivated X chromosome[25]. Somatic cells with two or more punctate H3K27me3 in nuclei were considered as polyploid cells. Currently, flow cytometry was the major way to evaluate the polyploid stromal cells. Here, we established a visible method to estimate polyploidization, which will greatly advance mechanisms to study polyploidy.

Previously studies suggested that endoreduplication and the absence of cytokinesis were the primary reasons for polyploidization of stromal cells which were mediated by various cyclin-associated proteins[3,5,27]. Here, we found that cytokinesis program was significantly enriched in *Men1*^d/d decidual cells. We also provided a previously unappreciated mechanism for this black box. *Cdkn1b* (encoding p27), a recognized Menin-MLL1/2 target gene[13], was decreased in *Men1*^d/d uteri. Furthermore, multiple observations proved that female mice lacking functional p27 were infertile[52–54]. Furthermore, based on the observation that p27 induced polyploidy of liver cells without cytokinesis[32,55], we speculated p27 also participated in regulating decidual cell polyploidization. Though the mRNA of *Cdkn1b* was reduced, the H3K4me3 modification at the promoter of *Cdkn1b* between *Men1*^f/f and *Men1*^d/d stroma was comparable, indicating an alternative regulation of Menin on *Cdkn1b* which deserved further investigation.

Collectively, our results provided a perspective that the patterning of decidual cells into disparate fates through the concerted activity of BMP and FGF signaling pathways underpinned the progression of decidualization. It's known that the gland is also critical for the decidualization processing as reported in the Ltf-Cre mediated FOXA2 deletion mouse model[23]. In this study, we mainly investigated the role of *Men1* in decidualization based on both in vivo and in vitro evidence. While how *Men1* regulates glands growth and distribution in decidualized stroma and the physiological role of glandular *Men1* in decidualization deserve further investigation. Results reported here revealed that the blunt induction of BMP2 caused by unrestrained FGF2-ERK1/2 signaling activity deranged the development of SDZ due to disorganized genomic H3K4me3 distribution in the absence of *Men1*. PTX3, an extra-cellular trap for FGF2, was identified as a critical Menin-H3K4me3 regulated downstream gene in decidual regionalization and pregnancy maintenance by curbing the cross-talk between FGF2-ERK1/2 and BMP2. Our findings will remarkably advance our knowledge on current model of decidualization and inspire future studies for human fertility improvement.

## Methods

**Animals**. Two-month-old C57BL/6 male and female mice were used in the present study. *Men1* floxed mouse line (*Men1*^f/f) was constructed as previously described[10]. Uterine-specific knockout mice (*Men1*^d/d) were generated by mating *Men1*^f/f females with *Pgr*^cre/+ males[16]. Mice were housed in the animal care facility of Xiamen University with a controlled environment (22 ± 2 °C, 50–60% humidity, 12-h light/dark cycle, lights on at 7 AM) and free access to food and water according to the guidelines for the care and use of laboratory animals. All experimental procedures were approved by the Animal Welfare Committee of Research Organization (X200811), Xiamen University.

**Analysis of pregnancy events**. Two-month-old female mice were mated with fertile wild-type males to induce pregnancy (vaginal plug = day 1 of pregnancy). Plug-positive females were kept separately for pregnant experiments. Pregnancy rate and litter size were monitored for the whole pregnancy process. For day 5 and day 6 pregnant mice, implantation sites were visualized by an intravenous injection of 100 μL of 1% Chicago blue in saline. The number and average weight of implantation sites demarcated by distinct blue bands was recorded. Uterine horns were flushed to check for the presence of embryos if no blue band was observed. Mouse blood samples were collected on day 6 and serum E2, as well as P4 levels, were measured by radioimmunoassay. At least three mice were used for every individual experiment in each mouse model.

**Artificial decidualization model**. Pseudopregnant mice were obtained by mating 2-month-old females with vasectomized males of the same strain. To induce artificial decidualization, one uterine horn of pseudopregnant mice was infused with sesame oil (25 μl) on day 4; the non-infused contralateral horn was taken as a control. The weight of infused and non-infused uterine horns was recorded 4 days after the oil infusion and the fold increase in uterine weights was served as an index of decidualization[24].

**In situ hybridization**. In situ hybridization with isotopes or digoxygenin (DIG) was modified according to the previously described method[24]. Frozen sections (10 μm) were mounted onto poly-L-lysine coated slides and stored at −80 °C until used. After removal from −80 °C, the slides were placed on a slide warmer (37 °C) for 3 min and then fixed in 4% paraformaldehyde in PBS for 15 min at 4 °C. Following prehybridization, uterine sections were hybridized to specific cRNA probes overnight at 65 °C. After hybridization, the slides were incubated with RNase A (10 mg/ml) at 37 °C for 30 min. RNase A-resistant hybrids were detected by autoradiography using liquid emulsion or Anti-Digoxigenin-AP, Fab fragments (Roche). Mouse-specific cRNA probes labeled with isotope or digoxin for *Men1*,

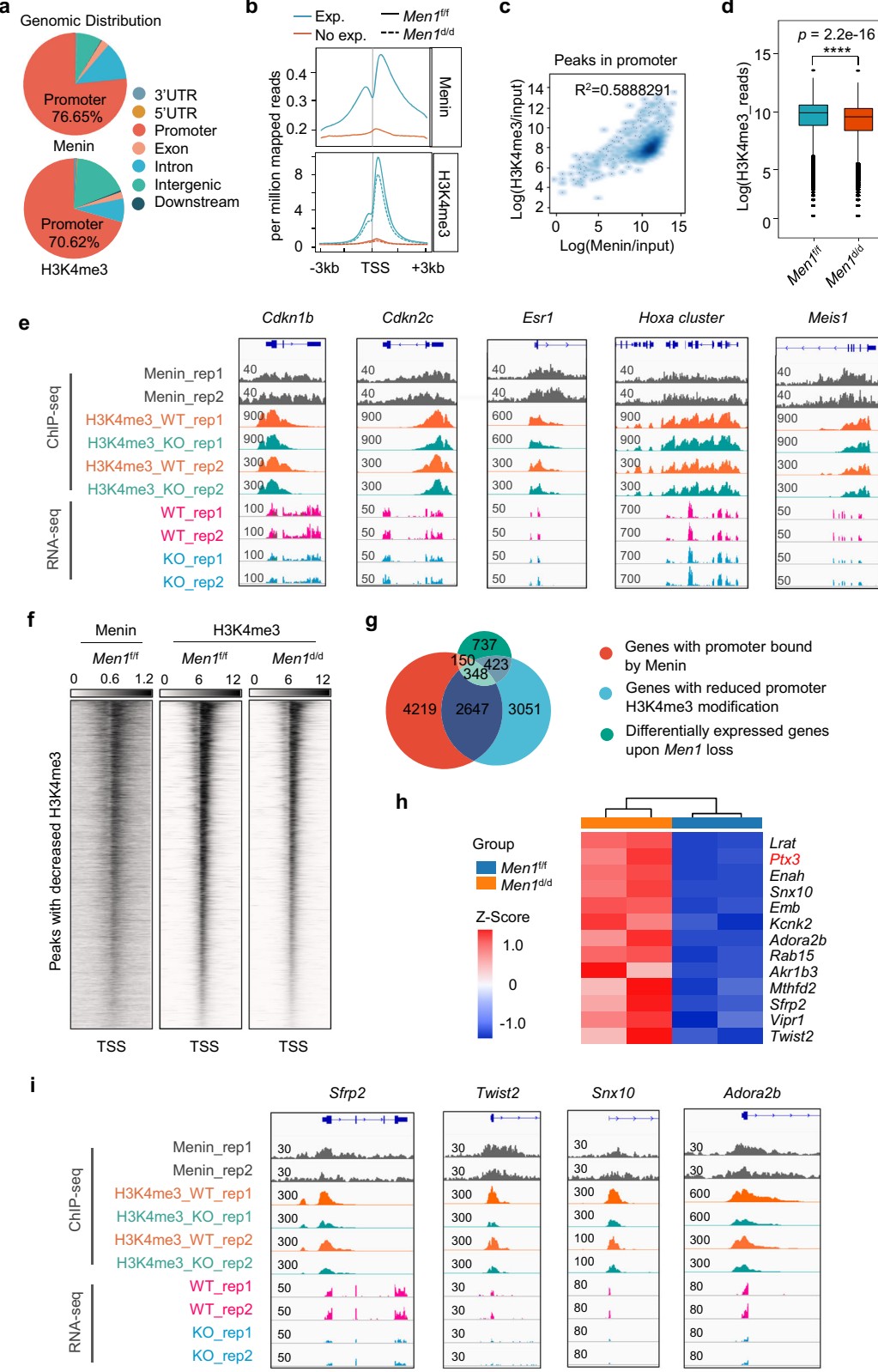

Bmp2, Dan, Igfbp5, Bmp7, Wnt4, Prl8a2, Hoxa10, Ptn, Sfrp4, Ptx3, and Fgfr1 were used for hybridization.

**Immunostaining.** The tissue specimens were fixed overnight in 10% neutral buffered formalin and then dehydrated in increasing concentrations of ethyl alcohol, followed by clearing of alcohol by xylene. Uterine slices were deparaffinized and incubated in citrate buffer for antigen retrieval by hyperbaric heating and then incubated overnight at 4 °C with the primary antibodies including Menin (Bethyl, 1:1000), COX2 (Santa Cruz, 1:200), Dtprp (homemade, 1:200), Ki67 (Service-bio,1:200), PL1 (Santa Cruz, 1:200), 3β-HSD (Santa Cruz, 1:200), p450scc (Santa Cruz, 1:200), PR (Cell Signaling Technology, 1:200), HAND2 (Santa Cruz, 1:200), p27 (Abcam,1:100), pH3 (Cell Signaling Technology, 1:200), ERK1/2 (Cell Signaling Technology, 1:200) and p-ERK1/2 (Cell Signaling Technology, 1:200). A

**Fig. 6 H3K4me3 levels are decreased in a limited number of genes downregulated in *Men1*-deficient uteri. a** Genomic distribution of Menin and H3K4me3 peaks in decidual tissues on day 8. **b** Menin and H3K4me3 ChIP-seq signal density at TSS. TSS, transcriptional start site. **c** Correlation between Menin and H3K4me3 ChIP-seq signal in gene promoters. $R^2 = 0.5888291$. **d** Change of H3K4me3 reads the number at gene promoters between *Men1*f/f and *Men1*d/d decidual tissues on day 8. ****$p = 2.2e-16$. The boxplots where midlines indicate medians, boxes indicate the interquartile range and whiskers indicate minimum/maximum range, and the statistical significance was calculated by the Wilcoxon test. **e** Genome browser view of normalized Menin and H3K4me3 ChIP-seq signals and RNA-seq tracks for known target genes in *Men1*f/f and *Men1*d/d decidual tissues on day 8. **f** Coverage profiles for Menin and H3K4me3. Heatmap of peaks with decreased H3K4me3 following *Men1* deletion centered at TSS. TSS transcriptional start site. **g** Venn diagram showing the overlap between genes with reduction H3K4me3 in promoters by *Men1* loss ($n = 6,469$), genes with promoter bound by Menin ($n = 7364$) and genes differentially expressed upon *Men1* loss ($n = 1659$). **h** Heatmap of top-ranked genes that are most likely to be the targets of Menin-H3K4me3. RPKM of each gene from different samples was normalized to Z-score. **i** Genome browser view of normalized ChIP-seq signals and RNA-seq tracks for Menin-H3K4me3 target genes in *Men1*f/f and *Men1*d/d decidual tissues.

Histostain-SP Kit (Zhongshan Golden Bridge Biotechnology) was applied to visualize the antigen. Immunofluorescence staining for OCT4 (Cell Signaling Technology, 1:200), β-Catenin (Abcam, 1:200), H3K27me3 (Cell Signaling Technology, 1:200), BrdU (Abcam, 1:500), PCNA (Santa Cruz, 1:200) and FOXA2 (Abcam, 1:500) was performed in paraffin-fixed sections and secondary antibody Cy™3 AffiniPure Goat Anti-Rabbit IgG(H + L) (Jackson ImmunoResearch,1:200) were used. The images were captured by Leica DM2500 light microscope. Antibodies with detailed information are listed in Supplementary Table 1.

**Western blot**. Decidual cells were isolated as described previously[17]. The proteins were separated by using 10% SDS-PAGE and transferred to polyvinylidene difluoride membranes. And then the membranes with proteins were blocked with 5% skim milk in TBST for 1 h at room temperature (RT) and incubated overnight at 4 °C with primary antibody for Menin (Bethyl, 1:2000), β-Actin (Bioworld, 1:3000), Dtprp (homemade, 1:2000), ERK1/2 (Cell Signaling Technology, 1:1000), p-ERK1/2 (Cell Signaling Technology, 1:1000), BMP2 (Abcam, 1:500), SMAD1/5/8 (Cell Signaling Technology, 1:1000), p-SMAD1/5/8 (Cell Signaling Technology, 1:1000), H3K4me3 (Abcam, 1:2000) and H3 (Abmart, 1:2000). After incubation with primary antibody, membranes were incubated with specific secondary antibodies (Zhongshan Golden Bridge Biotechnology, 1:5000) for 2 h at RT. Bands were visualized using Supersignal West Pico (Thermo Scientific) according to the manufacturer's instructions. Uncropped and unprocessed scans of the blots are presented in the Supplementary Fig. 13. Antibodies with detailed information are listed in Supplementary Table 1.

**Quantitative real-time PCR**. Total RNA was extracted from uterine tissues or cells using TRIzol reagent (Invitrogen) following the manufacturer's protocol. About 1 μg RNA was used to synthesize cDNA. Expression levels of the genes were validated by quantitative real-time PCR analysis with SYBR Green (TAKARA). All assays were performed at least three times. All PCR primers are listed in Supplementary Table 2.

**Flow cytometry**. Day 8 decidual cells were digested and harvested. The cell pellet was suspended in 0.25 ml PBS after centrifugation; 1 ml of cold 80% ethanol was added dropwise under constant and gentle vortexing. Samples were incubated for 30 min on ice and subsequently overnight at −20 °C before being subjected to staining. Cell sediments were suspended in staining solution (PBS containing 5 mg/ml Propidium Iodide and 2 mg/ml DNase-free RNase A) the next day. Samples were incubated for 30 min at 37 °C in the dark. They were then returned to RT and subjected to flow cytometry using Beckman Cytoflex. Flow cytometry data were analyzed using FlowJo software (v10, Tree star). The intensity value of SSC-A and FSC-A of fluorescence-activated cell sorting (FACS) are used for loose gates. The intensity value of FSC-A and FSC-H of FACS are used to gate a single cell. The polyploid cells are distinguished by the intensity value of FSC-A which is proportional to cell size and the intensity value of Propidium iodide staining which is used to measure DNA content. The experiments were repeated three times.

**RNA-seq**. Total RNA was extracted from day 8 decidua (dissect out the embryo and muscle layer) of *Men1*f/f and littermates *Men1*d/d using TRIzol reagent (Invitrogen) according to the manufacturer's protocol. Purified RNA was prepared and subjected to RNA sequencing using BGISEQ-500 platform (China, BGI). RNA-seq raw data were initially filtered to obtain clean data after quality control by Trimgalore. High-quality clean data were aligned to the mouse reference genome (mm10) using STAR.

**RNA-Seq data analysis**. Differential expression genes were normalized to fragments per kilobase of exon model per million mapped reads (RPKM) using the EdgeR3.9 package in R with the criteria of fold change significantly greater than 1.5 and $p < 0.05$. The visualization of RNA-Seq data was done by ggplot2 package in R. Gene ontology (GO) enrichment analysis of differential expression genes (DEGs) were analyzed by using DAVID (the Database for Annotation, Visualization, and Integrated Discovery, [https://david.ncifcrf.gov]). GO terms with corrected $p < 0.05$ were considered significantly enriched. Gene Set Enrichment Analysis (GSEA) was performed to determine whether a defined set of genes shows statistically significant differences between *Men1*f/f and littermates *Men1*d/d decidual tissues.

**ChIP-Seq**. Chromatin immunoprecipitation (ChIP) was modified and performed according to the previously described standard protocol[56]. Decidual tissue wetted with PBS quickly cut into small pieces with an iris scissor. Snipped tissue was fixed with 1% formaldehyde (Cell Signaling Technology) for 10 mins at RT (for H3K4me3 ChIP), or fixed with disuccinimidyl glutarate (DSG, Santa Cruz, 2 mM) for 30 min at RT and then double fixed with 1% formaldehyde for another 10 mins at RT (for Menin ChIP). Fixation was stopped by glycine (0.125 M). Tissue homogenate obtained by using Dounce Tissue Grinder was filtered through a 200 μm cell strainer to remove connective tissue. Fixed cells were lysed by Lysis Buffer 1 (50 mM HEPES pH 7.5, 1 mM EDTA, 140 mM NaCl, 0.5% NP-40, 10% glycerol, 0.25% Triton X-100), Lysis Buffer 2 (10 mM Tris-HCl pH8.0, 1 mM EDTA, 0.5 mM EGTA, 200 mM NaCl) and Lysis Buffer 3 (10 mM Tris-HCl pH8.0, 1 mM EDTA, 0.5 mM EGTA, 100 mM NaCl, 0.1% Sodium Deoxycholate, 0.1% N-lauroylsarcosine). All lysis buffers should be supplemented with protease inhibitors cocktail (Roche) before use. Chromatin DNA was sheared to an average 300–500 bp by using a BioRuptor sonicator (Diagenode). Solubilized chromatin was centrifuged to remove debris and incubated overnight at 4 °C with antibody Menin (Bethyl, 1:50) and H3K4me3 (Abcam, 1:50) bound to 20 μl protein A magnetic beads (Invitrogen). After washing and elution, the protein-DNA complex was reversed by heating at 65 °C overnight. Immunoprecipitated DNA was purified by using QIAquick spin columns (Qiagen). Immunoprecipitated and input DNA were quantified using Qubit 4.0 fluorometer. ChIP-seq libraries were prepared by using the KAPA DNA Hyper Prep Kit (KK8502) following the manufacturer's instruction and sequenced with an Illumina Nova PE150. Two biological replicates were performed for ChIP-seq. For ChIP-qPCR, chromatin was sheared by sonication until the average length of 500-1000 bp and pulled down by antibody Menin (Bethyl, 1:50), H3K4me3 (Abcam, 1:50) and Rabbit IgG (Cell Signaling Technology, 1:50) bound with 20 μl protein A magnetic beads (Invitrogen). Specific primers were used to detect immunoprecipitated chromatin fragments, as well as input chromatin. All PCR primers are listed in Supplementary Table 2.

**Primary uterine stromal cell culture**. Three to four pseudopregnant day 4 mouse uterine horns were cut into small pieces (2–3 mm). Tissue pieces were first digested in 3 ml fresh medium (HBSS antibiotic; Gibco) containing 6 mg/ml dispase (Gibco) and 25 mg/ml pancreatin (Sigma), and then incubated in fresh medium containing 0.5 mg/ml collagenase (Sigma) at 37 °C for 30 min. The digested cells were passed through a 70 μm filter to obtain the stromal cells. Cells were plated at 60 mm dishes or 6-wells plates, containing phenol red-free Dulbecco modified Eagle medium (DMEM) and Ham F12 nutrient mixture (1:1) (Gibco) with 10% charcoal-stripped fetal bovine serum (CS-FBS) and antibiotic. Two hours later, the medium was replaced with fresh medium (DMEM/F12, 1:1) with 10% CS-FBS. The next morning, the medium was replaced with DMEM/F12 containing 1% C-FBS, E2 (Sigma, 10 nM), P4 (Sigma, 1 μM) and antibiotic to induce decidualization. The media was changed every 48 h. For the treatment of primary uterine stromal cells, MI-503 (2 μM; MedChemExpress), recombinant murine FGF2 (PeproTech, 20 ng/ml), recombinant mouse PTX3 (R&D systems, 100 ng/ml) and MEK inhibitor PD0325901 (Selleck, 5 μM) were used.

**Statistical analysis**. All data are presented as mean ± SEM. Each experiment included at least three independent samples. Comparison between two groups was made by two-tailed unpaired Student's *t*-test. $P < 0.05$ was considered to indicate a significant result.

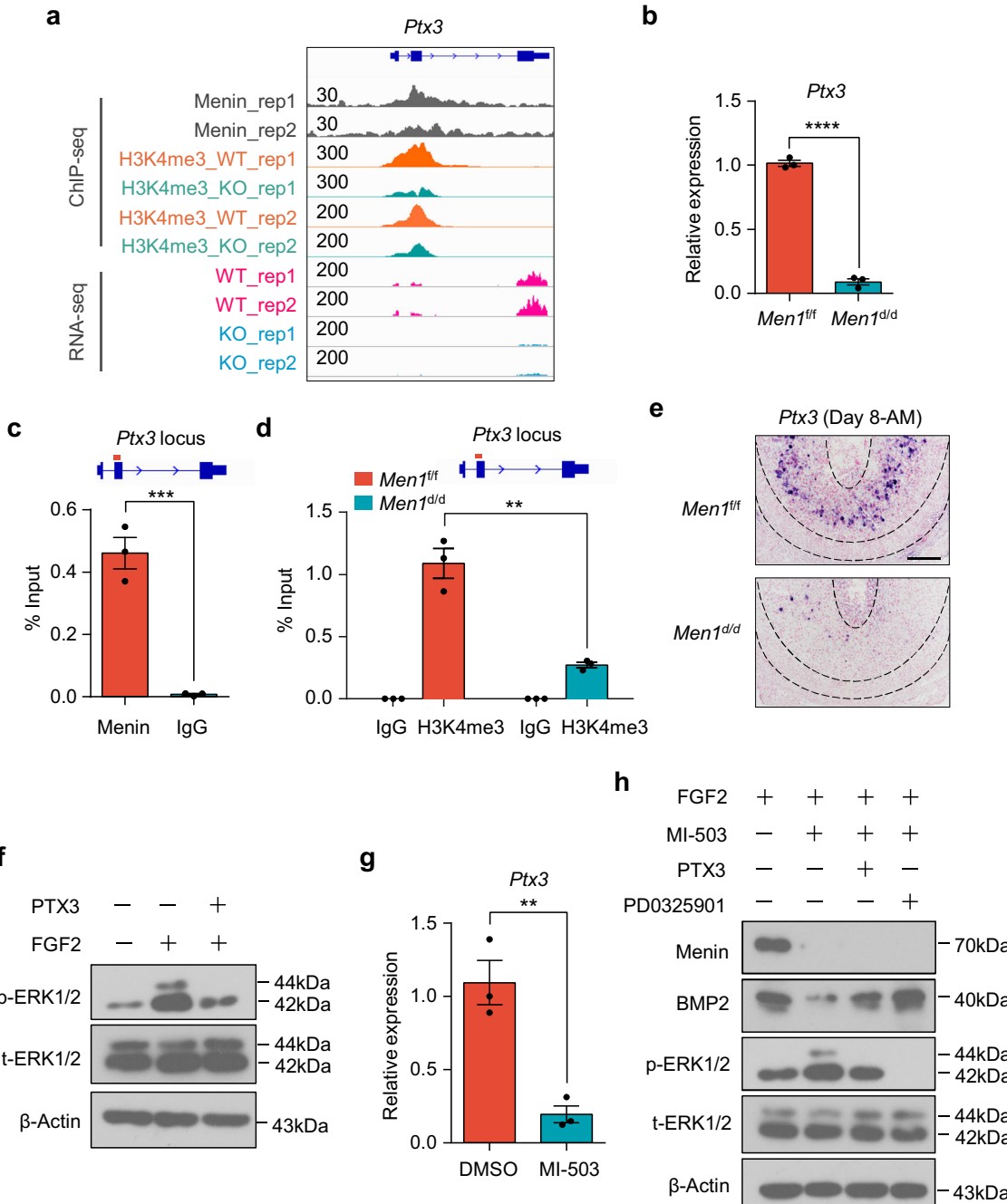

**Fig. 7 Menin inhibits the cross-talk between FGF2 and BMP2 by promoting *Ptx3* expression in an H3K4me3 dependent manner. a** Genome browser view of normalized Menin and H3K4me3 ChIP-seq signals at *Ptx3* promoter locus and RNA-seq tracks for *Ptx3* in *Men1*^f/f and *Men1*^d/d decidual tissues. **b** Quantitative real-time PCR analysis of *Ptx3* in *Men1*^f/f and *Men1*^d/d uteri on day 8. The values are normalized to *Gapdh* and indicated as the mean ± SEM ($n = 3$ biologically independent samples). Two-tailed unpaired Student's *t*-test, ****$p = 1.07e-5$. **c** Quantitative ChIP analysis of Menin at *Ptx3* promoter in uterine decidual tissue. Data represent the mean ± SEM. Two-tailed unpaired Student's *t*-test, ***$p = 0.0009$. **d** Quantitative ChIP analysis of H3K4me3 at *Ptx3* promoter in *Men1*^f/f and *Men1*^d/d uterine decidual tissue. Data represent the mean ± SEM ($n = 3$ biologically independent samples). Two-tailed unpaired Student's *t*-test, **$p = 0.0025$. **e** In situ hybridization of *Ptx3* in *Men1*^f/f and *Men1*^d/d uteri on day 8. Scale bar: 200 μm. **f** Immunoblotting analysis of the phosphorylation of ERK1/2 in mouse primary endometrial stromal cells treatment with FGF2 (20 ng/ml) in the absence or presence of PTX3 (100 ng/ml). Total ERK1/2 and β-Actin were used as loading controls. **g** Quantitative real-time PCR analysis of *Ptx3* in primary mESC treated with DMSO or MI-503(2 μM) in vitro. The values are normalized to *Gapdh* and indicated as the mean ± SEM ($n = 3$ biologically independent samples). Two-tailed unpaired Student's *t*-test, **$p = 0.0053$. **h** Immunoblotting analysis of Menin, BMP2, and phosphorylated ERK1/2 after the addition of PTX3 (100 ng/ml) or PD0325901 (5 μM) in mESC treatment with MI-503 (2 μM) in the presence of FGF2 (20 ng/ml). Total ERK1/2 and β-Actin were used as loading controls.

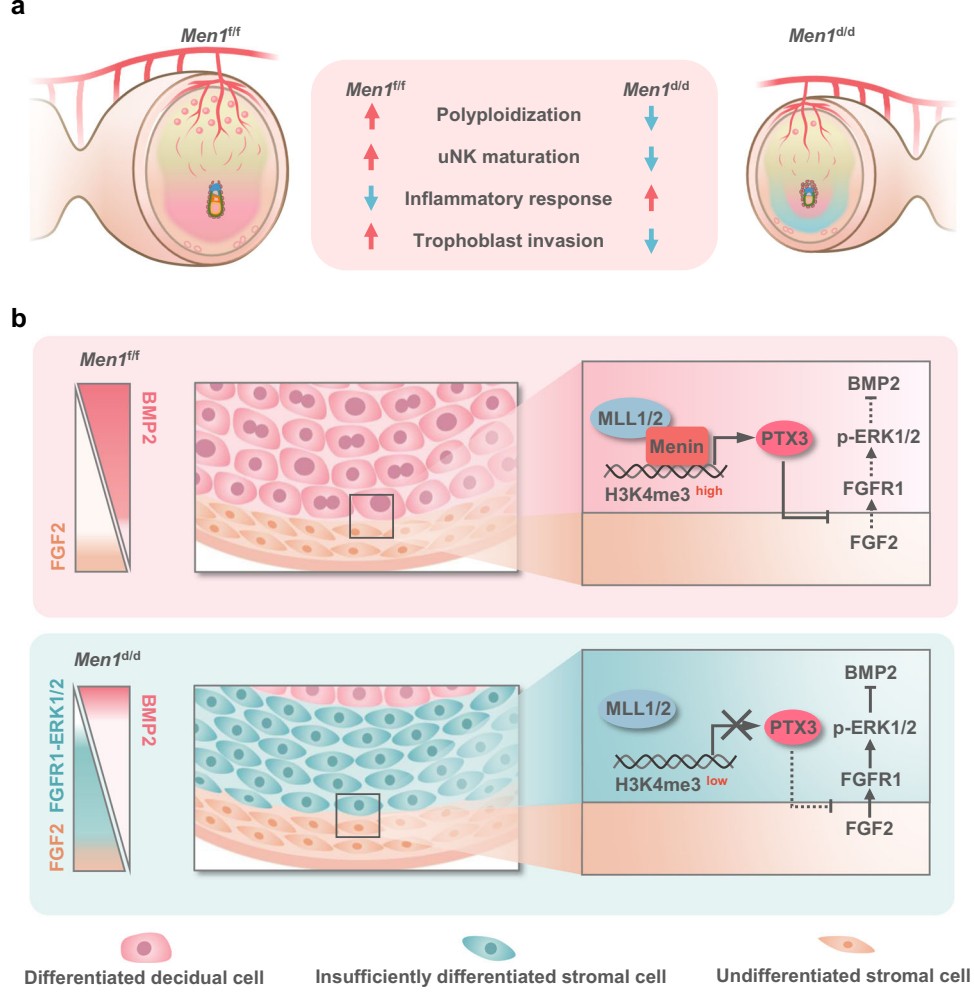

**Fig. 8 A proposed schematic diagram showing the function of uterine *Men1* during decidualization. a** Uterine deletion of *Men1* leads to abnormal polyploidization of decidual cells in the SDZ, accompanied by impeded maturation of uNK cells, increased inflammatory response, and shallow trophoblast invasion. **b** Menin positively regulates the expression of *Ptx3* in an H3K4me3 dependent manner. PTX3, trapping FGF2 from undifferentiated stromal cells, ensures the expression of BMP2 in the SDZ by inhibiting the activation of FGF2-ERK1/2.

**Reporting summary**. Further information on research design is available in the Nature Research Reporting Summary linked to this article.

## Data availability

The sequencing data generated in this study have been deposited in the Gene Expression Omnibus database under accession code GSE182525, GSE191327, and GSE182539. Source data are provided with this paper.

## Code availability

The data analysis pipeline used in this paper is deposited at [https://github.com/dwb0211/Menin].

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

## Acknowledgements

We are grateful to Prof. Francesco DeMayo (National Institute of Environmental Health Sciences) for providing us with the *Pgr*^Cre/+ mice. Thanks to Bo He and Yufei Jiang in the lab for animal model establishment. Element for the mouse in Supplementary Fig. 5 was created with BioRender. This work was supported by the National Key R&D Program of China (2021YFC2700302 to H.W., 2018YFC1004400 to S.K., 2017YFC1001402 to H.W., 2018YFC1004100 to W.D.), National Natural Science Foundation of China (31970797 to Z.L., 81830045 and 82030040 to H.W., 81971388 to S.K, 81701483 and 81971419 to W.D.), Fundamental Research Funds for the Central Universities (20720190073 to W.D.).

## Author contributions

Mengy.L., W.D., L.T., M.L., and C.G. performed experiments and prepared figures. Mengy.L., W.D., J.L., S.K., H.W., and Z.L. designed experiments. C.Z. provided the *Men1* loxP mouse model. Mengy.L., W.D., and S.K. analyzed data. Mengy.L., H.B., W.D., S.K., H.W., and Z.L. wrote the manuscript.

## Competing interests

The authors declare no competing interests.
