## [Peer Review File · Nature Communications]

Menin directs regionalized decidual transformation through epigenetically setting PTX3 to balance FGF and BMP signalingReviewers' Comments:

Reviewer #1:

Remarks to the Author:

This paper from Liu and colleagues identifies a previously unreported function for the Menin1 protein (Men1) in regulating development of the placenta in mouse. Overall this work is thorough and will be of significant impact to the developmental biology community. After performing a literature search I was unable to find any previous papers reporting a role for Men1 in the placenta, therefore I feel this manuscript meets the benchmark for impact and novelty for publication in Nature Communications. All of the mouse experiments are carefully performed with relevant controls and quantitation where appropriate. The authors provide a plausible mechanism for the developmental phenotype that they observe which is downregulated expression of the PTX3 gene (an inhibitor of FGF2 signaling). This in turn causes over-activation of the MAPK/ERK signaling pathway and results in an imbalance between proliferation and differentiation ultimately leading to disrupted placental architecture. Overall I am in favor in publication of this manuscript, however, I do have some criticisms that should be addressed.

1. In figure 6 I found the reductions in H3K4me3 to be modest and best and do not feel that they support the model that the changes in gene expression that accompany them are H3K4me3-dependent as the authors claim in the figure title. These ChIP-seq experiments were also performed without spike in normalization using *Drosophila* chromatin which makes the data difficult to interpret. In my experience the differences that they observe may simply be due to experimental variation in ChIP-seq assays. It is also worth noting that the authors observe a global drop of H3K4me3 at all expressed genes (Figure 6B) whereas global levels of H3K4me3 detected by western blotting are not dramatically altered (Supp Figure 6D). This further suggests that some of the differences the authors detect in H3K4me3 ChIP-seq may be experimental noise.

Previous experiments with Men1 knockout cells have found that H3K4me3 is not globally reduced in ChIP-seq but rather occurs at select loci that experience dramatic loss of H3K4me3 ("Genome-Wide Characterization of Menin-Dependent H3K4me3 Reveals a Specific Role for Menin in the Regulation of Genes Implicated in MEN1-Like Tumors" PLOS One 2012)

Another example of this can be seen in Figure 7B where PTX3 expression by RNA-seq is reduced almost to background whereas H3K4me3 is still easily detectable.

One way to address this would be to repeat the ChIP-seq experiments using multiple replicates and chromatin spike in normalization. Another approach would be to use ChIP Q-PCR to more accurately determine the degree of reduction at some of the Men1-dependent promoters that the authors identify. Alternatively, the authors could simply modify the text to downplay the role of H3K4me3 in this process.

2. The mass spectrometry data presented in figure 6D presents a number of proteins represented by only 1 peptide (low confidence hits). This also appears to be a hand selected list of proteins as opposed to a full list of all detected proteins. In my opinion, this data adds very little to the paper since it only confirms that Men1 is in the canonical MLL1/2 complex. For this reason I would suggest that the authors consider removing it due to the low coverage of the hits, or alternatively include a full list of all detected proteins in the supplementary data.

3. The authors should check figure the labelling of Figure 6B. On top of the figure it seems to indicate that it's comparing the difference in H3K4me3 levels between expressed/non-expressed genes in wildtype and Men1 mutant mice, however the labelling is for Men1(f/f). Overall I found the labelling for this figure to be very confusing and neither the figure legend nor the manuscript text clarified exactly what the plot was displaying.

Reviewer #2:
Remarks to the Author:
General Comments

Understanding the underlying mechanisms of decidualization in women is vital to improving women's reproductive health and pregnancy outcomes. The manuscript by Liu et al. utilizes a multi-omics approach to investigate the underlying causes of pregnancy loss in a mouse model lacking uterine expression of Men1. Within the present studies, the authors demonstrate that Men1 mediates decidual progression and pregnancy success through the regulation of Ptx3 expression. Overall, this is a scientifically sound manuscript that goes well beyond phenotype observation by identifying the potential underlying mechanism of decidualization failure and pregnancy loss in the absence of uterine Men1. However, the paper will be strengthened by addressing the points detailed below.

Specific comments:

- 1.) To support the claim that Men1 is intensely expressed in proliferating stromal cells (lines 113-115) colocalization for Men1 and either Ki67 or pHH3 should be performed. Quantification of the results from this experiment should be included.
- 2.) Authors should include whole implantation site images for the IHC presented in figure 1 to compare protein and transcript expression directly. For this reviewer, it is challenging to interpret lines 120-122 without the corresponding images.
- 3.) The authors concluded that mice lacking Men1 had normal initiation of decidualization based on Bmp2 and Dan ISH. However, BMP2 transcript abundance and circumference around the implanting embryo appear to be reduced. This may result from variability between implantation sites; still the authors should select more representative images or provide quantification.
- 4.) Analysis of the progression of in vitro decidualization should be included in the Menin inhibitor study presented in supplemental figure 7. This can be done by measuring the expression of decidualization-related genes or by western blot for Dtprp.
- 5.) Based on the histology presented in supplemental figures 1A and 2D it appears that glandular epithelial cells are present on the mesometrial side of the uterus. The progesterone receptor cre driver results in recombination during uterine development and could result in a uterine developmental defect. Was this observation common in the mutant mice? The authors should stain for Foxa2 to ensure that glands have developed only on the antimesometrial side of the uterus. The potential of a uterine developmental defect should be addressed in the discussion
- 6.) Figure 5e: Although the p-smad1/5/8 band from men1d/d mice appears to be reduced in the presented blot, quantification of band intensity normalized to total smad1/5/8 would be helpful. In the same regard, quantification is necessary for supplemental figure 7a.
- 7.) Figure 6h: The top of "737" in the Venn diagram is cutoff, and "348" is larger font size.

Responses to Reviewer 1

Reviewer #1 (Remarks to the Author):

This paper from Liu and colleagues identifies a previously unreported function for the Menin1 protein (Men1) in regulating development of the placenta in mouse. Overall this work is thorough and will be of significant impact to the developmental biology community. After performing a literature search I was unable to find any previous papers reporting a role for Men1 in the placenta, therefore I feel this manuscript meets the benchmark for impact and novelty for publication in Nature Communications. All of the mouse experiments are carefully performed with relevant controls and quantitation where appropriate. The authors provide a plausible mechanism for the developmental phenotype that they observe which is downregulated expression of the PTX3 gene (an inhibitor of FGF2 signaling). This in turn causes over-activation of the MAPK/ERK signaling pathway and results in an imbalance between proliferation and differentiation ultimately leading to disrupted placental architecture. Overall I am in favor in publication of this manuscript, however, I do have some criticisms that should be addressed.

Response: We are pleased with this reviewer's comments and we also sincerely thank his/her valuable feedback to our manuscript. We have supplemented extra data to make our results convincing according to the reviewers' comments. Our responses to his/her comments are addressed below.

Question 1: *In figure 6 I found the reductions in H3K4me3 to be modest and best and do not feel that they support the model that the changes in gene expression that accompany them are H3K4me3-dependent as the authors claim in the figure title. These ChIP-seq experiments were also performed without spike in normalization using Drosophila chromatin which makes the data difficult to interpret. In my experience the differences that they observe may simply be due to experimental variation in ChIP-seq assays. It is also worth noting that the authors observe a global drop of H3K4me3 at all expressed genes (Figure 6B) whereas global levels of H3K4me3 detected by western blotting are not dramatically altered (Supp Figure 6D). This further suggests that some of the differences the authors detect in H3K4me3 ChIP-seq may be experimental noise.*

Previous experiments with Men1 knockout cells have found that H3K4me3 is not globally

reduced in ChIP-seq but rather occurs at select loci that experience dramatic loss of H3K4me3 ("Genome-Wide Characterization of Menin-Dependent H3K4me3 Reveals a Specific Role for Menin in the Regulation of Genes Implicated in MEN1-Like Tumors" PLOS One 2012)

Another example of this can be seen in Figure 7B where PTX3 expression by RNA-seq is reduced almost to background whereas H3K4me3 is still easily detectable.

One way to address this would be to repeat the ChIP-seq experiments using multiple replicates and chromatin spike in normalization. Another approach would be to use ChIP Q-PCR to more accurately determine the degree of reduction at some of the Men1-dependent promoters that the authors identify. Alternatively, the authors could simply modify the text to downplay the role of H3K4me3 in this process.

Response: We sincerely appreciate the reviewer's concerns and constructive suggestions to strength our manuscript. To exclude the experimental variation in ChIP-seq assays, we repeated H3K4me3 ChIP-Seq and reanalyzed the data using ChIPseqSpikelnFree normalization method (**Bioinformatics** 2020 Feb 15;36(4):1270-1272). Both experiments showed that H3K4m3 enrichment around the TSS regions of coding genes was modest decreased in *Men1* deficient cells (**See below Figure 1, only for reviewer**). Specifically, significantly reduced H3K4me3 accumulation in Menin targeted genes (e.g. *Ptx3*, *Sfrp2*, *Twist2*, *Snx10* and *Adora2b*) were observed in both experiments, accompanying with the reduced mRNA level of these genes (**Revised Fig.6i and 7a**).

As an indispensable component of MLL1/2 HMT complex that catalyzes H3K4me3 in a locus-specific manner, Menin is usually associated with transcriptional activation of genes. We and other studies revealed that the global H3K4me3 was not significantly affected in the absence of *Men1* (**Revised Supplementary Fig. 9a**) (**PLoS One** 2012;7(5):e37952; **Cell Rep** 2017 Mar 7;18(10):2359-2372; **Mol Cancer Res** 2015 Apr;13(4):689-98). There are several possibilities for this observation. First, the reduced H3K4me3 is mainly observed in TSSs in a limited number of downregulated genes in the absence of *Men1*, but the 789 upregulated genes have an increased H3K4me3 (**See below Figure 2, only for reviewer**), indicating the increased deposited H3K4me3 in upregulated genes that would be catalyzed by other HMT like SET1A/B and MLL3/4 may have a compensatory effect for the decreased H3K4me3 in the absence of Menin. Secondary, although the H3K4me3 levels across regions proximal to TSSs of a subset of genes were reduced in the absence of *Men1*, it is possible that the remaining vast

unchanged H3K4me3 in whole genome including the non-promoter region contribute to the unobvious change of gross H3K4me3 in *Men1* deficient mice. Lastly, the decidua is highly heterogeneous encompassing stromal cells, endothelial cells, immune cells and glands. The intact *Men1* in PR negative cells would also be attributed to this observation.

To avoid overstatement of our results per the reviewer's suggestion, we changed the title of Fig. 6 to "H3K4me3 levels are decreased in a limited number of genes downregulated in *Men1*-deficient uteri."

As suggested, we also repeated the ChIP qPCR assay for *Ptx3* as well as conducted new ChIP qPCR assays for the above target genes and confirmed that the promoter of the *Men1* target genes we identified have an unequivocal reduced H3K4me3 modification upon uterine *Men1* deletion (**Revised Fig.7d and Supplementary Fig. 10a**).

Figure 1: H3K4me3 ChIP-seq signal density around regions proximal to TSSs was modestly reduced in the absence of *Men1* in both biological replicates. TSS, transcriptional start site

promoter of upregulated genes

Figure 2: Increased H3K4me3 reads number at promoter of upregulated genes upon uterine *Men1* deletion on day 8 decidual tissues. $p=0.0023$.

Question 2: *The mass spectrometry data presented in figure 6D presents a number of proteins represented by only 1 peptide (low confidence hits). This also appears to be a hand selected list of proteins as opposed to a full list of all detected proteins. In my opinion, this data adds very little to the paper since it only confirms that Men1 is in the canonical MLL1/2 complex. For this reason I would suggest that the authors consider removing it due to the low coverage of the hits, or alternatively include a full list of all detected proteins in the supplementary data.*

Response: Thanks for this comment to strength our manuscript. We totally agree with the reviewer's suggestion. Based on the reviewer's suggestion, we have removed the part of mass spectrometry data in the revised manuscript after careful consideration.

Question 3: *The authors should check figure the labelling of Figure 6B. On top of the figure it seems to indicate that it's comparing the difference in H3K4me3 levels between expressed/non-expressed genes in wildtype and Men1 mutant mice, however the labelling is for Men1(f/f). Overall I found the labelling for this figure to be very confusing and neither the figure legend nor the manuscript text clarified exactly what the plot was displaying.*

Response: We sincerely appreciate this concern. We feel sorry for our negligence and

have corrected it in our revised manuscript to make the figure legend and manuscript text more clear (**Revised Fig. 6b**).

Responses to Reviewer 2

Reviewer #2 (Remarks to the Author):

General Comments

Understanding the underlying mechanisms of decidualization in women is vital to improving women's reproductive health and pregnancy outcomes. The manuscript by Liu et al. utilizes a multi-omics approach to investigate the underlying causes of pregnancy loss in a mouse model lacking uterine expression of Men1. Within the present studies, the authors demonstrate that Men1 mediates decidual progression and pregnancy success through the regulation of Ptx3 expression. Overall, this is a scientifically sound manuscript that goes well beyond phenotype observation by identifying the potential underlying mechanism of decidualization failure and pregnancy loss in the absence of uterine Men1. However, the paper will be strengthened by addressing the points detailed below.

Response: While we are pleased with the positive comments of this reviewer, we also appreciate his/her suggestions to strengthen the manuscript. We have addressed all of the suggestions in the revised manuscript and have performed additional experiments as advised.

Specific comments:

Question 1: *To support the claim that Men1 is intensely expressed in proliferating stromal cells (lines 113-115) colocalization for Men1 and either Ki67 or pHH3 should be performed. Quantification of the results from this experiment should be included.*

Response: Many thanks for this constructive suggestion to strength our manuscript. As suggested, we conducted new experiments to examine the colocalization for Menin and Ki67 on day 6 of pregnancy. Fluorescence staining of Ki67 indicated that stromal cells surrounding the embryo had ceased proliferation and undergo differentiation to form the PDZ on day 6. While stromal cells adjacent to the PDZ were still proliferating (**Revised Supplementary Fig. 1a**). The expression intensity of Menin in Ki67 positive stromal cells outside the PDZ was significantly higher than that in differentiated Ki67 negative stromal cells in the PDZ (**Revised**

Supplementary Fig. 1a, b).

Question 2: *Authors should include whole implantation site images for the IHC presented in figure 1 to compare protein and transcript expression directly. For this reviewer, it is challenging to interpret lines 120-122 without the corresponding images.*

Response: Thanks for the reviewer's concern. As the reviewer suggested, we have provided a lower magnification image of Menin staining, which include the cross section for whole uteri or implantation sites on days 4, 5, 6 and 8 of pregnancy (**Revised Fig. 1b**). As shown in revised Fig.1b, localization of Menin protein is broader than that of *Men1* mRNA expression in mouse uteri.

Question 3: *The authors concluded that mice lacking *Men1* had normal initiation of decidualization based on *Bmp2* and *Dan* ISH. However, *BMP2* transcript abundance and circumference around the implanting embryo appear to be reduced. This may result from variability between implantation sites; still the authors should select more representative images or provide quantification.*

Response: Thanks for this suggestion to strength our conclusion. We have replaced the controversial image with a more representative one of *Bmp2* staining (**Revised Fig. 2d**). In addition, to make our conclusion that mice lacking *Men1* had no significant influence on *Bmp2* expression on day 5 more convincing, we also conducted quantitative real-time PCR to determine the level of *Bmp2* transcript and found no significant difference of *Bmp2* mRNA expression between *Men1*^{f/f} and *Men1*^{d/d} uteri (**Revised Supplementary Fig. 2b**). Thus, we concluded that mice lacking *Men1* had normal initiation of decidualization.

Question 4: *Analysis of the progression of in vitro decidualization should be included in the Menin inhibitor study presented in supplemental figure 7. This can be done by measuring the expression of decidualization-related genes or by western blot for *Dtprp*.*

Response: Thanks for reviewer's concern. As suggested, we provided evidence that although the expression of Menin was significantly decreased by MI-503 treatment in the progression of decidualization at both 48h and 72h, the protein expression of *Dtprp* was not affected in the absence of FGF2, same for the decidualization-related genes *BMP2* (**Revised**

Supplementary Fig. 11b, e). This is different from the scenario of decidualization with the FGF2 treatment (**Revised Fig. 7h**), which display the decreased BMP2 expression and defective decidualization, highlighting the significance of FGF2 for regulating BMP2 expression and appropriate decidualization.

Question 5: *Based on the histology presented in supplemental figures 1A and 2D it appears that glandular epithelial cells are present on the mesometrial side of the uterus. The progesterone receptor cre driver results in recombination during uterine development and could result in a uterine developmental defect. Was this observation common in the mutant mice? The authors should stain for Foxa2 to ensure that glands have developed only on the antimesometrial side of the uterus. The potential of a uterine developmental defect should be addressed in the discussion.*

Response: We sincerely appreciate the reviewer's concern and constructive suggestion. We carefully examined the gland distribution in prepubertal, pre-implantation and post-implantation uteri.

We noticed that glands in the lateral areas near the mesometrial pole were increased in the *Men1* deletion uterus as characterized by a well-recognized gland marker FOXA2 (**Proc Natl Acad Sci U S A** 2017 Feb 7;114(6):E1018-E1026) (**Revised Supplementary Fig. 4**). Interestingly, the abnormal gland growth in *Men1^{d/d}* uteri occurred in adulthood, rather than the prepubertal stage with the initiation of gland genesis (**See below Figure 3a, only for reviewer**), which was probably contributed by the excessive proliferation of glands by circulating E2 as evidenced by increased Ki67 in day 4 glands (**See below Figure 3b, only for reviewer**).

In this study, we mainly investigated the role of *Men1* in decidualization based on both in vivo and in vitro evidence. While how *Men1* regulates glands growth after puberty and the physiological role of glandular *Men1* in decidualization is out the scope of current study and deserve further investigation. We provide the gland developmental data in both genotypes and discuss the potential cause of this defect in the revised manuscript.

Figure 3: (a) Immunohistochemistry of FOXA2 in *Men1^{fl/fl}* and *Men1^{d/d}* uteri on PND 20 and PND 60. PND, postnatal day. Scale bar: 200 μ m. (c) Immunohistochemistry of Ki67 in *Men1^{fl/fl}* and *Men1^{d/d}* uteri on pregnant day 4. Scale bar: 200 μ m.

Question 6: *Figure 5e: Although the p-smad1/5/8 band from men1d/d mice appears to be reduced in the presented blot, quantification of band intensity normalized to total smad1/5/8 would be helpful. In the same regard, quantification is necessary for supplemental figure 7a.*

Response: Thanks for this concern. We have provided quantitative analysis of p-SMAD1/5/8 band intensity normalized to total SMAD1/5/8 (**See revised Supplementary Fig. 8**). The quantification of Menin, BMP2 and Dtprp in the progression of in vitro decidualization was shown in **Supplementary Fig. 10b** in the revised version.

Question 7: *Figure 6h: The top of "737" in the Venn diagram is cutoff, and "348" is larger font size.*

Response: We appreciate this concern. We have carefully amended the mistakes in the revised manuscript (**Revised Fig. 6h**).

We again appreciate your time and efforts in handling this manuscript. We hope that the revised manuscript is satisfactory to you and the reviewers for consideration of publication in **Nature**

Communications.

Sincerely,

Haibin Wang

Haibin Wang, PhD., Distinguished Professor

Reproductive Medical Center, The First Affiliated Hospital of Xiamen University, Medical College,

Xiamen University, Xiamen, Fujian 361102, PR China. Tel: 86-592-2880501;

Email: haibin.wang@vip.163.com

Reviewers' Comments:

Reviewer #1:

Remarks to the Author:

The authors have adequately addressed my comments in the revised manuscript.

Reviewer #2:

Remarks to the Author:

The authors have addressed all my comments and the manuscript is suitable for publication.

Responses to Reviewer

Reviewer #1 (Remarks to the Author):

The authors have adequately addressed my comments in the revised manuscript.

We thank this reviewer for his/her positive statement.

Reviewer #2 (Remarks to the Author):

The authors have addressed all my comments and the manuscript is suitable for publication.

We are grateful for the reviewer's positive evaluation.